# MARSHAL: Incentivizing Multi-Agent Reasoning via Self-Play with Strategic LLMs

**Huining Yuan**[1*], **Zelai Xu**[2*], **Zheyue Tan**[3], **Xiangmin Yi**[1],
**Mo Guang**[4] , **Kaiwen Long**[4], **Haojia Hui**[4], **Boxun Li**[5], **Xinlei Chen**[1], **Bo Zhao**[3],
**Xiao-Ping Zhang**[1†], **Chao Yu**[1†], **Yu Wang**[2†]
[1]SIGS, Tsinghua University, [2]EE, Tsinghua University, [3]Aalto University,
[4]Li Auto Inc., [5]Infinigence AI

🌐 Project Page   🔧 Code   🤗 Models

## Abstract

Developing Large Language Models (LLMs) to cooperate and compete effectively within multi-agent systems (MASs) is a critical step towards more advanced intelligence. While reinforcement learning (RL) has proven effective for enhancing reasoning in single-agent tasks, its extension to multi-turn, multi-agent scenarios remains underexplored due to the challenges of long-horizon credit assignment and agent-specific advantage estimation. To address these challenges, we introduce **MARSHAL**, an end-to-end RL framework that incentivizes Multi-Agent Reasoning through Self-play witH strAtegic LLMs in both cooperative and competitive games. MARSHAL features a turn-level advantage estimator that aligns learning signals with each interaction for credit assignment, and an agent-specific advantage normalization to stabilize multi-agent training. By learning with self-play across cooperative and competitive games, MARSHAL agents trained from Qwen3-4B develop strong strategic abilities, with up to 28.7% performance improvements in held-out games. More importantly, the capability acquired through self-play generalizes beyond games, yielding consistent performance gains of MASs in reasoning benchmarks. When integrated into leading MASs, our MARSHAL agent achieves significant zero-shot performance gains of up to 10.0% on AIME, 7.6% on GPQA-Diamond, and 3.5% on average across all benchmarks. These results establish self-play in strategic games as a powerful approach for developing generalizable multi-agent reasoning capabilities in LLMs.

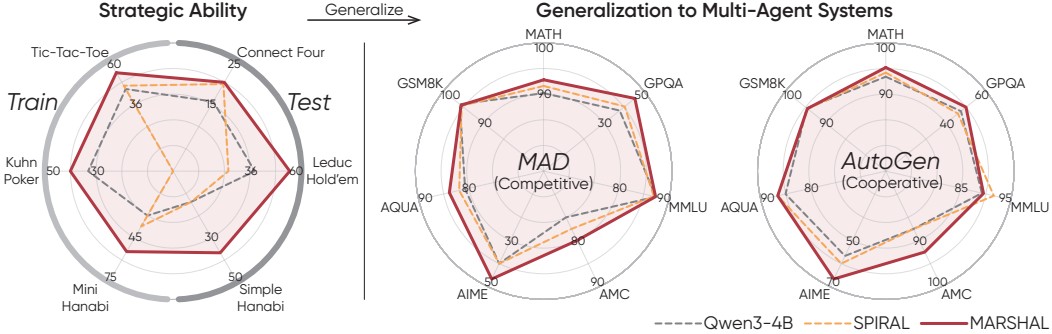

Figure 1: Evaluation of MARSHAL and two baselines on strategic games and reasoning benchmarks. MARSHAL incentivizes multi-agent reasoning ability via self-play in strategic games and generalizes to improvements of multi-agent systems like MAD and AutoGen on math and QA benchmarks.

---

*Equal contribution: {yuanhuining0, zelai.eecs}@gmail.com

†Corresponding authors: xpzhang@ieee.org, yuchao@sz.tsinghua.edu.cn, yu-wang@tsinghua.edu.cn

# 1 INTRODUCTION

The remarkable capabilities of Large Language Models (LLMs) have revolutionized numerous domains, enabling strong performance on a wide range of tasks from question answering to code generation (Achiam et al., 2023; Team et al., 2023). However, many real-world scenarios, such as negotiation (Bianchi et al., 2024), strategic gameplay (Silver et al., 2016; FAIR et al., 2022), and collaborative software development (Qian et al., 2023; Wu et al., 2024a) inherently involve multiple agents interacting over long horizons. Enabling LLMs to cooperate and compete effectively within multi-agent systems (MASs) is a critical frontier for advancing artificial intelligence.

While reinforcement learning (RL) has demonstrated remarkable success in enhancing the reasoning capabilities of individual LLMs (Guo et al., 2025; Team et al., 2025), its extension to multi-turn, multi-agent tasks faces two critical challenges. First, the problem of long-horizon credit assignment arises in multi-turn interactions. As a sequence of actions can each result in an immediate reward while collectively leading to the final sparse reward, accurately attributing the contribution of the action in each turn is inherently more complex than standard single-turn settings. Second, multi-agent training couples heterogeneous game roles with asymmetric information and different payoff scales, which introduces variance in advantage estimation, destabilizing the training process.

In this work, we present **MARSHAL** (Multi-Agent Reasoning through Self-play witH strAtegic LLMs), an end-to-end RL framework that incentivizes multi-agent reasoning capabilities via self-play in strategic games. We introduce two novel techniques to address the challenges in multi-turn, multi-agent self-play with Group-Relative Policy Optimization (GRPO) (Shao et al., 2024). First, we propose a simple yet effective turn-level advantage estimator to enable fine-grained credit assignment. This allows the model to accurately attribute long-term outcomes to individual actions and provide learning signals across multiple turns and agents. Second, we propose an agent-specific advantage normalization that stabilizes the training process by calibrating advantage estimates relative to the performance of each agent. This normalization accounts for the heterogeneous roles in multi-agent systems and ensures stable policy updates. By integrating these components, MARSHAL enables robust self-play learning from game outcomes, empowering LLMs to develop strategic abilities and multi-agent reasoning skills through cooperation and competition with themselves.

To evaluate the performance and generalization ability of MARSHAL agents, we conduct extensive experiments by training Qwen3-4B in a diverse range of cooperative and competitive games. Specifically, MARSHAL agents exhibit strong strategic ability across all game environments, with up to 28.7% performance improvements in three held-out games. More importantly, the capability acquired through self-play in games further generalizes to improvements of multi-agent systems in reasoning benchmarks. When integrated into both cooperative and competitive multi-agent systems, including AutoGen (Wu et al., 2024a) and MAD (Liang et al., 2023), our MARSHAL agents achieve zero-shot performance improvements of up to 10.0% on AIME, 7.6% on GPQA, and 3.5% on average across all benchmarks. We further conduct ablation studies to validate the effectiveness of our algorithmic design, complemented by a comprehensive analysis of reasoning patterns and failure modes to understand the successful generalization. The evaluation results establish MARSHAL as a powerful approach for developing generalizable multi-agent reasoning capabilities in LLMs.

In summary, our contributions are as follows:

- We propose MARSHAL, an end-to-end RL framework that enhances multi-agent reasoning through self-play in a diverse range of strategic games.
- We introduce two novel techniques of turn-level advantage estimator and agent-specific normalization to address the credit assignment and advantage variance in multi-turn, multi-agent RL training for LLMs.
- We perform extensive experiments and ablation studies to show that MARSHAL incentivize strong strategic ability and multi-agent reasoning capability that are generalizable to held-out games and multi-agent LLM systems.

# 2 PRELIMINARY

In standard RL fine-tuning for tasks like math, the environment is static. The model generates a complete Chain-of-Thought response, and only then is a terminal reward assigned. This process is

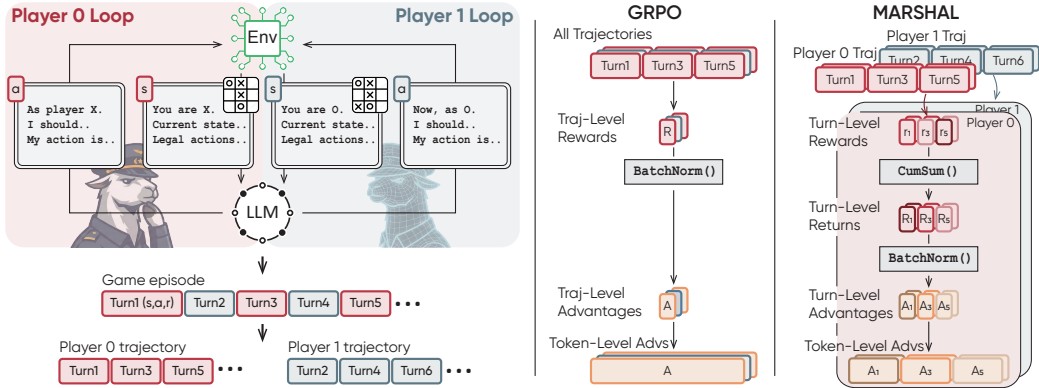

Figure 2: Overview of MARSHAL. Left column: generating player trajectories through self-play in strategic games. Middle column: naive advantage estimation by GRPO. Right column: advantage estimation by MARSHAL for accurate credit assignment in multi-turn, multi-agent setting.

typically modeled as a **token-level Markov decision process (MDP)**. Here, a trajectory consists of a single "turn", which is the generation of one complete response with multiple tokens. The state at step $t$ is the sequence of previously generated tokens $(o_1, ..., o_{t-1})$, and the action is the selection of the next token $o_t$. Given a task question $q$, the goal is to optimize the token-level policy $\pi(o_t|q, o_{<t})$ to produce a sequence that maximizes the final sparse reward.

In contrast, strategic games introduce a more complex, hierarchical decision-making structure. An entire game, not a single response, constitutes an episode. This is best modeled as a **turn-level MDP**, where decisions occur at two levels. At the high level, the state $s_k$ represents the state of the game at the beginning of turn $k$ (e.g., the board configuration, the cards in hand, etc.). A high-level action, $a_k$, corresponds to the agent's entire output for that turn (e.g., "I place my X in the top-left corner and here is why..."). This action $a_k$ is itself a sequence of multiple tokens generated by the LLM's low-level, autoregressive policy. In this case, the goal is to optimize the turn-level policy $\pi(a_k|s_k) = \prod_{t=1}^{T} \pi(o_{k,t}|s_k, o_{k,<t})$ to maximize the total return of all turns $R = \sum_{k=1}^{K} r_k$.

## 3 METHOD

In this section, we introduce **MARSHAL**, an end-to-end RL framework that enhances multi-agent reasoning ability of LLM through self-play in a diverse range of cooperative and competitive games. We begin by outlining the overall architecture of our training framework, building upon Group-Relative Policy Optimization (GRPO) (Shao et al., 2024). We then detail our primary technical contributions for addressing the credit assignment and advantage estimation challenge in multi-turn, multi-agent training. Finally, we describe the selection of our game environments and design of reward structures. An overview of our proposed method is shown in Fig. 2.

### 3.1 SELF-PLAY WITH GRPO

To eliminate the extensive computational cost introduce by the critic model in classic Proximal Policy Optimization (PPO) (Schulman et al., 2017), GRPO rollout each query for multiple times, constructing a group of $G$ responses $\{o^i\}_{i=1}^{G}$ and their corresponding outcome rewards $\mathbf{r} = \{r^i\}_{i=1}^{G}$, then estimate the advantage of each response by their relative reward to their corresponding group, which assigns equal advantage to the whole sequence (Fig.2 middle), yielding

$$\mathcal{J}_{\text{GRPO}}(\theta) = \mathbb{E}_{q \sim P(Q), \{o^i\}_{i=1}^{G} \sim \pi_{\theta_{old}}(O|q)} \frac{1}{G} \sum_{i=1}^{G} \frac{1}{|o^i|} \sum_{t=1}^{|o^i|} \mathcal{J}_{\text{surr}}(\pi_\theta; \pi_{\theta_{old}}, A_t^i, \varepsilon), \quad (1)$$

where $\mathcal{J}_{\text{surr}}(\pi_\theta; \pi_{\theta_{old}}, A_t^i, \varepsilon) = \min\left[\frac{\pi_\theta(o_t^i|q, o_{<t}^i)}{\pi_{\theta_{old}}(o_t^i|q, o_{<t}^i)} A_t^i, \text{clip}\left(\frac{\pi_\theta(o_t^i|q, o_{<t}^i)}{\pi_{\theta_{old}}(o_t^i|q, o_{<t}^i)}, 1 - \varepsilon, 1 + \varepsilon\right) A_t^i\right]$ is the token-level PPO surrogate objective, and the advantages are estimated by $A_t^i = \frac{r^i - \text{mean}(\mathbf{r})}{\text{std}(\mathbf{r})}$.

In multi-agent self-play, where all players in a strategic game is controlled by the same model, each episode of game results in a multi-turn trajectory for each player role, respectively. As a naive generalization of the original GRPO to our multi-turn scenario, we start by considering all trajectories from a game environment as a group of response, i.e. $\{(s_k^i, a_k^i)_{k=1}^{K^i}\}_{i=1}^{G}$, with total return as the terminal reward for each trajectory $\mathbf{r} = \{R^i\}_{i=1}^{G}$. Then, with an additional summation for multiple turns, GRPO directly generalized to

$$\mathcal{J}_{\text{GRPO}}^{\text{multi}}(\theta) = \mathbb{E}_{s_k^i \sim P(S), o_{k,t}^i \sim \pi_{\theta_{old}}(O|s_k^i, o_{k,<t}^i)} \frac{1}{G} \sum_{i=1}^{G} \frac{1}{K^i} \sum_{k=1}^{K^i} \frac{1}{|o_k^i|} \sum_{t=1}^{|o_k^i|} \mathcal{J}_{\text{surr}}(\pi_\theta; \pi_{\theta_{old}}, A_{k,t}^i, \varepsilon), \quad (2)$$

where the advantages are estimated by $A_{k,t}^i = \frac{R^i - \text{mean}(\mathbf{r})}{\text{std}(\mathbf{r})}$. This assigns equal advantage to all tokens in the multi-turn trajectory, similar to the original GRPO. For multi-game training, trajectories for each game are naturally considered separate groups and normalized independently.

## 3.2 ADVANTAGE ESTIMATION FOR MULTI-TURN, MULTI-AGENT LEARNING

While GRPO has demonstrated remarkable efficiency and stability in single-turn settings, such direct application to the multi-turn, multi-agent structure of self-play introduces significant challenges of long-horizon credit assignment and agent-specific advantage estimation. To address these issues, we introduce two novel modifications to GRPO.

**Turn-level advantage estimator.** To enable fine-grained credit assignment across a long trajectory, we incorporate the sequence of turn-level rewards $\mathbf{r} = \{(r_k^i)_{k=1}^{K}\}_{i=1}^{G}$. This setup is analogous to the "Process Supervision" setting in the original GRPO paper (Shao et al., 2024). In that setting, the proposed advantage estimation involves first normalizing all rewards across the entire batch, $\tilde{r}_k^i = (r_k^i - \text{mean}(\mathbf{r}))/\text{std}(\mathbf{r})$, and then computing the cumulative sum of these normalized values, $A_k^i = \sum_{\hat{k}=k}^{K} \tilde{r}_{\hat{k}}^i$. However, since the distributions of intermediate rewards can vary significantly, treating the entire set of rewards as a single distribution for global normalization is likely inappropriate and potentially problematic.

To address this problem, we propose a crucial inversion of these two steps: we **first sum, then normalize** (Fig.2 right). Specifically, we begin by calculating the standard Monte Carlo return (or cumulative reward) from each turn $k$ onwards: $R_k^i = \sum_{\hat{k}=k}^{K} r_k^i$. We then compute the advantage by normalizing these returns to their mean: $A_{k,t}^i = R_k^i - \text{mean}(\mathbf{R})$, where $\mathbf{R}$ is the collection of all cumulative rewards in the group. This formulation is equivalent to Generalized Advantage Estimation (GAE) (Schulman et al., 2015) with $\gamma = 1$ and $\lambda = 1$, where the value function $V(s_k)$ is approximated by a simple yet effective baseline: the empirical mean of the batch returns $\mathbb{E}[\mathbf{R}]$. By ensuring the final advantages are properly centered, this method provides a much more stable learning signal for multi-turn decision-making.

**Agent-specific advantage normalization.** In many games, the expected return can be highly dependent on a player's role (e.g., player 0 vs. player 1, or different roles in a cooperative game). Normalizing advantages across different roles would force all players toward a shared baseline, which is statistically inappropriate and can obscure role-specific learning signals.

To address this, we refine our advantage calculation further. We partition the batch of trajectories into sub-groups based on player role and apply the turn-level advantage estimator described above independently within each sub-group $G^p$ (Fig.2 right), where $p$ denotes the player index. This ensures that the advantage for a given action is calculated relative to the average outcome for that specific role, providing a more accurate and stable credit assignment in multi-agent settings. This agent-specific, sum-then-normalize advantage estimation forms the final objective for MARSHAL as

$$\mathcal{J}_{\text{MARSHAL}}(\theta) = \mathbb{E}_{s_k^{p,i} \sim P(S), o_{k,t}^{p,i} \sim \pi_{\theta_{old}}(O|s_k^{p,i}, o_{k,<t}^{p,i})}$$

$$\frac{1}{P} \sum_{p=1}^{P} \frac{1}{G_p} \sum_{i=1}^{G_p} \frac{1}{K^i} \sum_{k=1}^{K^i} \frac{1}{|o_k^{p,i}|} \sum_{t=1}^{|o_k^{p,i}|} \mathcal{J}_{\text{surr}}(\pi_\theta; \pi_{\theta_{old}}, A_{k,t}^{p,i}, \varepsilon), \quad (3)$$

where $A_{k,t}^{p,i} = R_k^{p,i} - \text{mean}(\mathbf{R}^p)$, $\mathbf{R}^p$ denotes the total set of cumulative rewards from subgroup $G^p$.

### 3.3 GAME ENVIRONMENTS

To incentivize comprehensive multi-agent reasoning ability for MARSHAL, we select a range of six strategic, two-player games. These games are partitioned into a training set and a more complex, held-out testing set for out-of-distribution (OOD) evaluation:

- **Perfect-information, competitive games:** To enable deterministic planning and role adaptation, we train on *Tic-Tac-Toe*, requiring the agent to recognize its strategic position (e.g., first-mover vs. second-mover) and plan accordingly. For evaluation, *Connect Four* serves as a more complex out-of-distribution test due to its vastly larger state space.
- **Imperfect-information, competitive games:** To foster robust reasoning and decision-making under imperfect information and uncertainty, we train on *Kuhn Poker*, a simplified poker variant. We evaluate generalization on the more sophisticated *Leduc Hold'em*.
- **Imperfect-information, cooperative games:** To develop social intelligence like intent recognition and Theory of Mind, we consider the classic cooperative card game *Hanabi*. Specifically, we train on *Mini Hanabi* and evaluate on a more challenging variant *Simple Hanabi* to test the generalization of cooperative strategies.

Collectively, this diverse range of both competitive and cooperative games ensures the LLM agent develops generalizable multi-agent reasoning capabilities.

### 3.4 REWARD DESIGN

While our primary learning signal is derived from the intrinsic game outcomes to minimize reward engineering, we incorporate two auxiliary rewards to ensure stable training. The final reward signal is composed of three components:

- **Intrinsic game rewards:** The primary signal is the intrinsic game outcome. This is a +/-1 reward for a win/loss/draw in *Tic-Tac-Toe*, chips won or lost in *Kuhn Poker* (max of 2), and a shared +1 reward per successfully played card in *Mini Hanabi* (max of 4). To balance these varying scales during multi-game training, we normalize the maximum reward across games to 4 by scaling the *Tic-Tac-Toe* reward with a factor of 2.
- **Format reward:** To ensure parsable outputs, we provide a small positive reward (+0.05) for validly formatted actions and a large negative penalty (-10.0) that terminates the game for invalid ones, similar to the approach in DeepSeek-R1 (Guo et al., 2025).
- **Response length penalty:** To encourage conciseness, we apply a turn-level penalty for verbosity, inspired by Kimi k1.5 (Team et al., 2025), which scales linearly for responses longer than a set threshold. The penalty is calculated as

$$r_{\text{length}}(l) = \alpha \cdot \max\left(0, 1 - \frac{l - l_{\min}}{l_{\max} - l_{\min}}\right),\tag{4}$$

where we set $l_{\min} = 11$, $l_{\max} = 2048$, and the scaling coefficient $\alpha = 0.5$.

## 4 EXPERIMENTS

In this section, we present extensive experiment results to validate MARSHAL's ability to obtain generalizable multi-agent reasoning capability. Additional results can be found in the Appendix.

### 4.1 EXPERIMENTAL SETUP

We use Qwen-4B (Yang et al., 2025), a state-of-the-art reasoning model, as our foundation to measure the multi-agent reasoning capabilities of MARSHAL. We train two model types: specialist agents on each game (*Tic-Tac-Toe*, *Kuhn Poker*, *Mini Hanabi*) and a generalist agent on all three simultaneously. Implementation details and hyperparameter settings are provided in Appendix B and C. Our primary baseline is SPIRAL, a recent work focused on self-play in purely competitive, zero-sum games (Liu et al., 2025). Our evaluation is structured in four stages (see Appendix D for details):

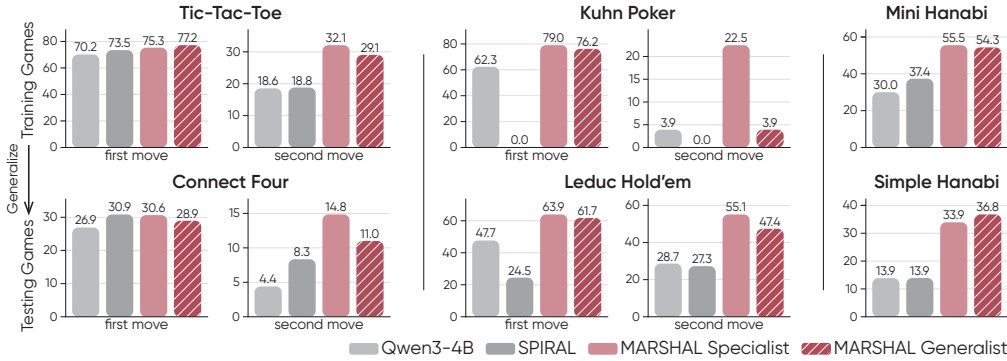

Figure 3: Average normalized game returns. Specialist agents not only master their training domains but also generalize effectively to their more complex, held-out counterparts (e.g., from *Tic-Tac-Toe* to *Connect Four*). The generalist model achieves consistently high performance across the entire suite of games, establishing it as the most robust and versatile agent.

1. **Strategic ability:** We first evaluate strategic competence by benchmarking agents against strong Monte Carlo Tree Search (MCTS) or Nash Equilibrium (NE) opponents. Performance is measured over 1000 games on both the training games and a suite of more complex, held-out games. For the cooperative *Hanabi* games, we report the standard self-play score.

2. **Generalization to multi-agent systems:** This is the primary test of this work. We integrate MARSHAL agents into two established systems: the competitive MAD framework (Liang et al., 2023) and the cooperative AutoGen framework (Wu et al., 2024a), measuring zero-shot generalization on standard math and QA benchmarks.

3. **Pattern analysis:** We conduct a qualitative analysis of the agent's reasoning process and a quantitative analysis of failure modes to understand the successful generalization to MASs.

4. **Ablation studies:** Finally, we perform ablation studies to validate our key algorithmic designs, particularly our novel advantage estimation technique.

## 4.2 STRATEGIC ABILITY

Our analysis begins with strategic ability on games, with normalized game return detailed in Fig 3. The MARSHAL framework proves highly effective: specialist agents not only master their training domains and outperform baselines, but also generalize effectively to more complex, held-out counterparts (e.g., from *Tic-Tac-Toe* to *Connect Four*). We also observe evidence of cross-category skill generalization. For example, the *Tic-Tac-Toe* specialist model demonstrates smooth improvement not only on *Tic-Tac-Toe*, but also the OOD *Mini Hanabi* (Fig. 4), suggesting MARSHAL cultivates foundational skills like turn-based planning that are broadly beneficial.

Crucially, the generalist model, trained on all environments simultaneously, achieves high performance across the entire suite of games, achieving 28.7% improvement on *Leduc Hold'em* and 22.9% on *Simple Hanabi*. Its broad competence across competitive and cooperative settings establishes it as our most robust agent overall.

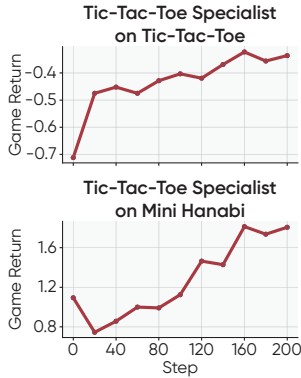

Figure 4: Eval curves of the *Tic-Tac-Toe* specialist in *Tic-Tac-Toe* and *Mini Hanabi*.

## 4.3 GENERALIZATION TO MULTI-AGENT SYSTEMS

The ultimate test of MARSHAL is whether skills honed in games generalize to practical, out-of-domain challenges. We evaluate this on a suite of demanding mathematics and QA benchmarks in a zero-shot manner, including MATH500 (Cobbe et al., 2021), GSM8K, AQUA-RAT (Ling et al., 2017), AIME24, AMC23, MMLU-STEM (Hendrycks et al., 2020), and GPQA-Diamond (Rein et al., 2024). As a preliminary step, we investigate how our game tasks enhance foundational reasoning in a

Table 1: Evaluation results on downstream reasoning benchmarks within multi-agent systems. Competitive game-trained agents excel in the competitive MAD framework, while the cooperative-trained agent excels in the cooperative AutoGen framework. The generalist model performs robustly across both. **Bold** and underlined indicate the best and second-best scores, respectively.

| Setting | Model | Average | Math | | | | | QA | |
|---|---|---|---|---|---|---|---|---|---|
| | | | MATH | GSM8K | AQUA | AIME | AMC | MMLU | GPQA |
| *Single Agent* | Qwen3-4B | 60.74 | 87.60 | 94.60 | 39.80 | 36.70 | 70.00 | 57.10 | **39.39** |
| | SPIRAL | **63.75** | 87.50 | 94.80 | 51.20 | 36.70 | **80.00** | 58.70 | 37.37 |
| | MARSHAL | | | | | | | | |
| | Tic-Tac-Toe | 63.54 | 89.10 | **95.20** | 46.50 | 40.00 | 77.50 | 57.60 | 38.89 |
| | Kuhn Poker | 61.38 | 87.80 | 94.50 | 48.40 | 33.30 | 72.50 | 59.30 | 33.84 |
| | Mini Hanabi | 62.05 | 88.10 | 94.70 | 48.00 | **43.30** | 65.00 | 58.90 | 36.36 |
| | **Generalist** | 62.79 | **89.90** | 94.60 | **52.00** | 33.30 | 75.00 | **59.90** | 34.85 |
| *MAD (Competitive)* | Qwen3-4B | 72.45 | 90.20 | 95.91 | 80.71 | 40.00 | 75.00 | **87.42** | 37.88 |
| | SPIRAL | 73.41 | 91.60 | 95.45 | 81.89 | 40.00 | 77.50 | 87.01 | 40.40 |
| | MARSHAL | | | | | | | | |
| | Tic-Tac-Toe | 75.01 | 92.20 | 96.06 | 83.07 | 43.33 | **82.50** | 86.76 | 41.12 |
| | Kuhn Poker | 74.54 | 91.60 | **96.21** | 82.68 | 40.00 | **82.50** | 87.39 | 41.41 |
| | Mini Hanabi | 73.70 | 91.40 | 95.60 | 82.68 | 43.33 | 77.50 | 87.04 | 38.38 |
| | **Generalist** | **75.96** | **92.80** | 95.60 | **83.86** | **46.67** | 80.00 | 87.36 | **45.45** |
| *AutoGen (Cooperative)* | Qwen3-4B | 79.14 | 93.40 | **94.69** | 85.04 | 56.67 | 87.50 | 89.21 | 47.47 |
| | SPIRAL | 80.05 | 94.20 | 94.47 | 86.61 | 60.00 | 87.50 | **91.60** | 45.96 |
| | MARSHAL | | | | | | | | |
| | Tic-Tac-Toe | 80.15 | 94.40 | **94.69** | **87.01** | 60.00 | 90.00 | 89.53 | 45.45 |
| | Kuhn Poker | 81.54 | **95.80** | 94.39 | 86.61 | 63.33 | 92.50 | 89.65 | 48.48 |
| | Mini Hanabi | 81.54 | 94.40 | 94.54 | 86.22 | **66.67** | **95.00** | 88.98 | 44.95 |
| | **Generalist** | **82.15** | 95.20 | 94.54 | 86.61 | **66.67** | 92.50 | 89.53 | **50.00** |

standard single-agent setting. Notably, MARSHAL models achieve notable improvements over both the Qwen-4B baseline on a number of math benchmarks, on par with SPIRAL (Table 1).

We further embed our agents into established multi-agent systems to directly measure their cooperative and competitive capabilities. In the competitive MAD debate framework, agents trained on competitive games show a clear advantage. Notably, the generalist agent is able to achieve an average of 3.51% over the original Qwen3-4B across all benchmarks. Conversely, in the cooperative AutoGen framework, the agents trained for cooperation—the *Hanabi* specialist and the generalist—excel across numerous benchmarks. In particular, the generalist model achieves a striking gain of 7.57% on GPQA-Diamond in the MAD framework, and 10.00% on AIME in the AutoGen framework.

These results provide compelling evidence that MARSHAL forges distinct, generalizable skills for both competition and collaboration. These capabilities directly translate to improved performance in downstream multi-agent applications, with the generalist model proving to be the most robust.

## 4.4 REASONING PATTERN AND FAILURE MODE ANALYSIS

To understand the mechanisms behind MARSHAL's successful generalization, we analyze the agent's reasoning (`<think>`) traces. This qualitative analysis reveals the emergence of key multi-agent reasoning patterns cultivated in our mix of strategic games. We highlight two representative patterns corresponding to the competitive and cooperative training environments, as showcased in Table 2.

First, reflecting the skills honed in competitive games, MARSHAL develops a role-aware strategy. In *Tic-Tac-Toe*, the agent explicitly identifies itself as the "second-move player" and adopts a defensive strategy. This skill generalizes directly to the MAD debate framework, where the agent recognizes its role as the "negative side assistant" and adopts a negative stance rather than just solving the problem.

More profoundly, MARSHAL cultivates intent recognition, a key component of Theory of Mind. In the cooperative *Hanabi*, the agent learns to interpret a teammate's actions as communications with hidden intent (e.g., "Maybe they want me to play this card?"). This sophisticated skill generalizes to the AutoGen framework, where the agent infers a collaborator's uncertainty from subtle cues, such

Table 2: Qualitative analysis of emergent reasoning patterns.

| Skill | Manifestation in Game-Play | Generalize to Multi-Agent Systems |
|---|---|---|
| Role Understanding | *The Tic-Tac-Toe specialist recognizes its role as the second player (O) and adopts a defensive strategy.*
`<think>`
Okay, so I am playing the game of Tic-Tac-Toe as the mark O... **As the second-move player, I should** prioritize blocking the X marks from forming a horizontal, vertical, or diagonal line...
`</think>` | *The same agent, acting as the "negative" debater in MAD, adapts its strategy to refute the opponent.*
`<think>`
Okay, so I need to figure out the minimum number of... The affirmative side just argued for 4. **As the negative side assistant, I should** try to find an alternative solution and convince the judge...
`</think>` |
| Intent Recognition | *The Hanabi specialist infers the intent behind a teammate's ambiguous hint.*
`<think>`
Okay, so I need to figure out the next move. Player 0 just revealed my only red card. Wait, but the rank remains unknown. **Maybe they want me to play this card to the stack?** ...
`</think>` | *The same agent, acting as a user proxy in AutoGen, infers uncertainty from a collaborator's missing 'TERMINATE' token.*
`<think>`
Okay, let's see. The assistant gave the answer 17.5 m/s, but did not end the conversation with 'TERMINATE'. **Maybe the assistant is not sure with this answer?** ...
`</think>` |

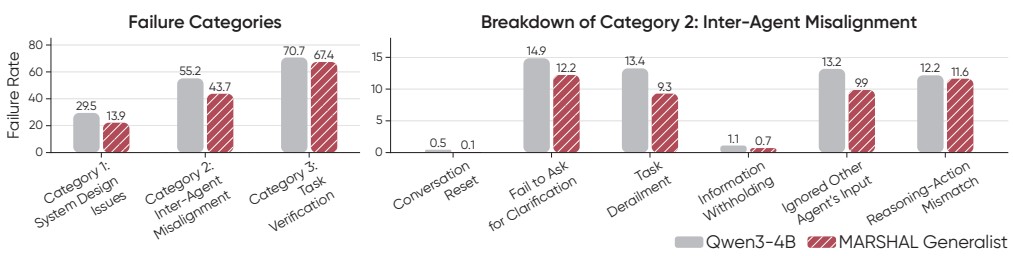

Figure 5: Percentage of different failure modes in GPQA-Dimond. Through MARSHAL training, the generalist agent significantly reduces the occurrence of Inter-Agent Misalignment.

as a missing TERMINATE token, instead of treating it as a simple error. Together, these patterns provide qualitative evidence that MARSHAL goes beyond improving benchmark scores; it equips the agent with the cognitive toolkit essential for effective multi-agent interaction.

To further substantiate these qualitative observations with quantitative evidence, we performed a failure mode analysis on the GPQA-Diamond benchmark within the MAD framework. Adopting the taxonomy of Cemri et al. (2025), we categorized failures into System Design Issues (e.g., formatting errors, loop repetition), Inter-Agent Misalignment (e.g., multi-agent reasoning failures like ignoring peers or task derailment), and Task Verification issues.

As shown in Fig 5 (left), while MARSHAL improves basic instruction following (reducing System Design Issues by ~7%), the reduction in Inter-Agent Misalignment is significantly larger (11.5%). To understand the drivers of this strategic improvement, we further decompose the Inter-Agent Misalignment category into its sub-categories. The breakdown reveals that the performance gains are primarily driven by reductions in Task Derailment and Ignored Other Agent's Input. This indicates that MARSHAL agent are actively listening to peers and maintaining focus on its objective, validating our hypothesis that game-theoretic self-play cultivates generalizable multi-agent reasoning skills.

## 4.5 ABLATION STUDIES

To validate our core algorithmic designs, we conduct two targeted ablation studies: (1) comparing self-play to training against a fixed opponent, and (2) sequentially removing the two key algorithmic components: our turn-level advantage estimator and agent-specific advantage normalization.

Table 3: Generalization comparison between MARSHAL (self-play) and its fixed-opponent variant. The latter exhibits significant overfitting to static environments and opponents. Values denote average normalized game returns; for competitive games, entries indicate *first-move / second-move* returns. Underlined scores indicate performance degradation compared to the standard MARSHAL model.

| Model | Training Games | | | Testing Games | | |
|---|---|---|---|---|---|---|
| | Tic-Tac-Toe | Kuhn Poker | Mini Hanabi | Connect Four | Leduc Hold'em | Simple Hanabi |
| MARSHAL (Tic-Tac-Toe) | 75.30 / 32.10 | 74.15 / 3.42 | 50.48 | 30.65 / 14.85 | 58.36 / 27.65 | 29.75 |
| *w/ fixed opponent* | 88.00 / 41.95 | 63.15 / 28.84 | 34.93 | 20.35 / 5.65 | 47.38 / 35.55 | 12.22 |
| MARSHAL (Kuhn Poker) | 69.85 / 25.50 | 79.04 / 22.49 | 44.98 | 27.60 / 12.70 | 63.94 / 62.10 | 29.35 |
| *w/ fixed opponent* | 0.00 / 0.00 | 76.19 / 15.64 | 0.00 | 0.00 / 0.00 | 0.00 / 0.00 | 0.00 |

Table 4: Ablation results for algorithmic design. Both our turn-level advantage estimator and agent-specific advantage normalization prove essential for performance. Notation follows Table 3.

| Model | Training Games | | | Testing Games | | |
|---|---|---|---|---|---|---|
| | Tic-Tac-Toe | Kuhn Poker | Mini Hanabi | Connect Four | Leduc Hold'em | Simple Hanabi |
| MARSHAL (Tic-Tac-Toe) | 75.30 / 32.10 | 74.15 / 3.42 | 50.48 | 30.65 / 14.85 | 58.36 / 27.65 | 29.75 |
| *w/o Turn-Level.* | 74.60 / 24.15 | 80.26 / 28.35 | 34.80 | 26.75 / 12.30 | 48.34 / 41.34 | 19.05 |
| *w/o Agent-Specific.* | 82.70 / 31.20 | 70.89 / 11.24 | 44.10 | 25.40 / 10.50 | 51.04 / 49.88 | 21.72 |
| MARSHAL (Kuhn Poker) | 69.85 / 25.50 | 79.04 / 22.49 | 44.98 | 27.60 / 12.70 | 63.94 / 62.10 | 29.35 |
| *w/o Turn-Level.* | 63.35 / 19.65 | 92.49 / 21.02 | 41.65 | 29.60 / 10.85 | 32.26 / 31.23 | 22.98 |
| *w/o Agent-Specific.* | 69.55 / 24.55 | 75.37 / 19.55 | 40.18 | 27.00 / 10.50 | 35.73 / 21.50 | 22.42 |
| MARSHAL (Hanabi) | 71.90 / 7.35 | 72.52 / 9.29 | 55.55 | 26.75 / 5.75 | 37.36 / 55.12 | 33.93 |
| *w/o Turn-Level.* | 67.55 / 10.60 | 68.45 / 31.78 | 53.20 | 25.25 / 3.05 | 54.79 / 47.77 | 30.68 |
| *w/o Agent-Specific.* | 68.15 / 13.40 | 74.15 / 10.27 | 52.50 | 32.10 / 5.10 | 44.30 / 56.41 | 32.08 |

**Self-play vs. fixed-opponent learning.** Training agents against fixed, expert opponents reveals an intuitive and critical flaw: the agents overfit. As shown in Table 3, these models fail to generalize, exhibitinga significant performance drop on games outside their direct training domain. This is particularly evident with the *Kuhn Poker* specialist, which suffers a severe degradation on non-poker games—a clear case of strategic mode collapse. These results confirm that the adaptive curriculum provided by self-play is essential for the development of robust, generalizable policies.

**Analysis of advantage estimation design.** We then ablate our two-part advantage estimation technique. First, reverting from our fine-grained turn-level advantage estimator to a coarse, trajectory-level reward signal causes a significant performance drop, especially for the *Tic-Tac-Toe* specialist, which is trained on a long-horizon game. Second, removing agent-specific advantage normalization also degrades performance, particularly in competitive games-trained model like the *Tic-Tac-Toe* specialist and the *Kuhn Poker* specialist where player experiences and agent return are distinct (Fig. 6). Conversely, the effect is mild in the *Hanabi* specialist, which is trained on a cooperative game with similar return distribution between players (Fig. 6). Together, these results confirm that both components of our advantage estimation are critical for effective learning in complex multi-turn, multi-agent environments.

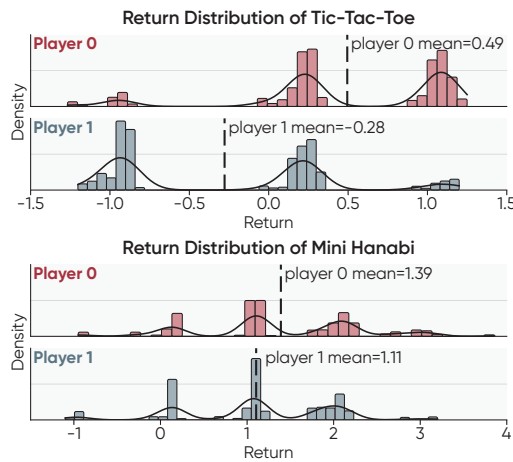

Figure 6: Return distribution of the generalist MARSHAL agent as different player roles on *Tic-Tac-Toe* and *Mini Hanabi*.

## 5 RELATED WORK

**LLM-based multi-agent systems.** To further extend the ability of LLMs with collaborative intelligence, LLM-based multi-agent systems have been proposed for various tasks. AutoGen (Wu et al., 2024a) and CAMEL (Li et al., 2023) design cooperative agents for general reasoning and question answering. MetaGPT (Hong et al., 2024) and ChatDev (Qian et al., 2023) propose static multi-agent workflows with specialized roles for software development tasks. Multi-Agent Debate (MAD) (Liang et al., 2023) takes a competitive approach by letting LLMs propose and criticize solutions over multiple rounds for the final answer. These works mainly focus on designing workflows for multi-agent interactions with fixed LLMs. Other works have proposed to generate workflows automatically (Zhang et al., 2024a). In comparison, our work takes a complementary approach by training LLMs to enhance their multi-agent reasoning capabilities and build stronger foundation models that can be integrated with these frameworks.

**Reinforcement learning for LLMs.** Reinforcement learning (RL) has emerged as a prominent approach to enhance LLMs' ability from instruction following to reasoning. To align LLMs with human values, RL from human feedback (RLHF) (Ouyang et al., 2022) and from AI feedback (RLAIF) (Bai et al., 2022) have been widely used in post-training to steer models towards favorable behaviors. The recent success of reasoning models (Jaech et al., 2024; Guo et al., 2025) further popularizes RL from verifiable rewards (RLVR) for incentivizing long Chain-of-Thought reasoning capability of LLMs and has been applied in mathematical reasoning (Shao et al., 2024), coding (Wei et al., 2025), and tool-using (Jin et al., 2025). However, these works primarily consider single-turn or single-agent tasks, while our approach extends RL training to multi-turn, multi-agent scenarios. Recently, MT-GRPO (Zeng et al., 2025) also address the turn-level credit assignment issue for multi-turn RL. However, they propose to normalize the turn-level rewards turn-by-turn, which can introduce variance in strategic games with varying numbers of turns. We propose a "sum-then-normalize" approach that takes advantage of all turns in the normalization, increasing robustness.

**Self-play training of LLMs.** The great success of achieving superhuman performance in multi-agent games (Silver et al., 2016; Vinyals et al., 2019; Berner et al., 2019) has spurred the use of self-play to train LLMs by playing with themselves (Zhang et al., 2024b). Some work (Chen et al., 2024; Wu et al., 2024b) combines the game-theoretic property of self-play to overcome the intransitivity of human preference in LLM alignment. Another line of work (FAIR et al., 2022; Xu et al., 2023; 2025a) uses self-play to improve the strategic ability of LLM agents in specific games. The most relevant work to ours is SPAG (Cheng et al., 2024) and SPIRAL (Liu et al., 2025), which also train LLMs via self-play to show generalization to reasoning tasks. However, this work focuses on zero-sum games and considers generalization in single-agent evaluations. In contrast, our work consider both cooperative and competitive games and demonstrate generalized improvement in multi-agent systems. As a concurrent work to ours, SPIRAL also addresses the heterogeneity of player roles in advantage estimation and introduces a Role-Conditioned Advantage Estimation (RAE), analogous to our agent-specific normalization. We advance this finding by revealing that role separation is only critical in games with distinct return distributions between players.

## 6 DISCUSSION

In this work, we introduced MARSHAL, a framework utilizing self-play in a diverse range of cooperative and competitive strategic games to cultivate multi-agent reasoning capabilities in LLMs. Our findings demonstrate that skills honed in strategic games directly translate to enhanced performance in general multi-agent systems, establishing self-play as a scalable paradigm for training LLM agents.

Despite these promising results, limitations remain. First, our study utilizes two-player games as efficient "prototypes" to cultivate foundational skills. While we demonstrate that these constrained settings are sufficient for strong generalization to standard reasoning tasks, scaling to larger $N$-player environments introduces significantly greater challenges regarding non-stationarity, population diversity, and credit assignment that warrant dedicated future investigation. Additionally, moving beyond classic games to complex "social sandboxes" (e.g., simulated software engineering, collaborative research) represents a compelling direction for training agents in more practical domains.

Ultimately, our work provides strong evidence that the principles of self-play are a powerful engine for progress, paving the way for the next generation of sophisticated LLM agents.

## REPRODUCIBILITY STATEMENT

All code, model checkpoints, and training scripts required to reproduce the findings of this paper are publicly available at https://github.com/thu-nics/MARSHAL. The repository includes all necessary configurations to replicate our key experiments.

## ETHICS STATEMENT

This work adheres to the ICLR Code of Ethics. The primary ethical consideration of our work is the dual-use nature of enhancing LLM agent capabilities. While our goal is to foster beneficial multi-agent reasoning skills like cooperation and competition, the resulting models could potentially be applied to malicious ends. We release our code and models to the research community to encourage further study into the safety and alignment of more capable agents.

## ACKNOWLEDGMENTS

This work is supported by the National Natural Science Foundation of China (No.62406159, 62325405), Ant Group, Beijing National Research Center for Information Science and Technology (BNRist), Beijing Innovation Center for Future Chips, Shenzhen Key Laboratory of Ubiquitous Data Enabling (No. ZDSYS20220527171406015), and Tsinghua Shenzhen International Graduate School-Shenzhen Pengrui Endowed Professorship Scheme of Shenzhen Pengrui Foundation.

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

## A    USE OF LLMS

In the preparation of this paper, LLMs were utilized as writing assistants. Their use were limited to improving grammar, clarity, and overall readability. The core research, including the methodology, experiments, and analysis, represents the original work of the authors.

## B    IMPLEMENTATION DETAILS

**Framework and software stack.**    Our implementation of MARSHAL is built upon ROLL (Wang et al., 2025), a robust open-source codebase for post-training LLMs with reinforcement learning. ROLL's native support for agentic, multi-turn rollouts provided a strong foundation for our system. To achieve high performance, ROLL leverages vLLM for efficient inference during the rollout phase (Kwon et al., 2023) and is built on Megatron-LM for distributed training (Shoeybi et al., 2019). All game environments used in this work were implemented using OpenSpiel (Lanctot et al., 2019) and VS-Bench (Xu et al., 2025b), ensuring the correctness and standardization of game logic.

**Training settings.**    We use Qwen-4B as our base model for all experiments reported in our main results. The training is conducted in a fully online manner, where self-play trajectories are immediately used for policy updates. Specialist agents are trained with a batch size of 128 trajectories, while the generalist agent uses a batch size of 384 (128 from each game). All models were trained for a total of 200 optimization steps. For optimization, we employ the Adam optimizer (Kingma, 2014) with a cosine annealing learning rate schedule, which warms up for 10 steps to a peak of $1 \times 10^{-6}$ before gradually decaying to 0.

**Generation parameters.**    During the self-play rollout phase, text is generated via nucleus sampling. We use a temperature of 0.6, a Top-P of 0.99, and a Top-K of 100. These parameters were selected to encourage a diverse yet coherent exploration of strategies.

**Hardware configuration.**    Models were trained on a single server with 8 NVIDIA H100 GPUs.

## C    HYPERPARAMETERS

To ensure reproducibility and consistency, we maintain a unified hyperparameter configuration for all trained models. The complete settings are provided in Table 5.

## D    EVALUATION DETAILS

**Strategic ability.**    To rigorously assess game-playing proficiency, we evaluate our agents against a suite of strong, fixed opponents. For competitive games, this includes:

- *Tic-Tac-Toe* and *Connect Four*: two MCTS agents with varying simulation counts (100/1000 for *Tic-Tac-Toe*, 10/100 for *Connect Four*) to test against different strengths.
- *Kuhn Poker*: The exact Nash Equilibrium (NE) policy.
- *Leduc Hold'em*: A NE policy approximated by $5 \times 10^8$ iterations of Counterfactual Regret Minimization (CFR).

In all competitive games, models are evaluated as both the first-move and second-move player, measured by the average game return. For the cooperative *Hanabi* variants, performance is measured by the standard self-play game return. In all game settings, agents are evaluated for 1000 games.

For the results on strategic ability reported in the main text, we normalize the scores with respect to the worst-observed performance and the theoretical optimal performance.

For the fixed-opponent ablation studies, the MCTS agent with 100 simulations is used for *Tic-Tac-Toe* as the opponent, and the NE policy is used for *Kuhn Poker*.

Table 5: Hyperparameters

| Category | Parameter | Value |
|---|---|---|
| Model Configuration | Max Response Length | 4096 |
| | Max Sequence Length | 32768 |
| | Dtype | bf16 |
| Training Settings | Max Steps | 200 |
| | Train Batch Size | 128 |
| | Optimizer | Adam |
| | Adam parameters ($\beta_1$, $\beta_2$) | (0.9, 0.95) |
| | Learning Scheduler | Cosine Annealing |
| | Learning Rate | $1 \times 10^{-6}$ |
| | Weight Decay | 0.05 |
| | Warm-up Step | 10 |
| | Gradient Norm Clip | 1.0 |
| RL Settings | Sampling Temperature | 0.5 |
| | PPO Epochs | 1 |
| | (top P, top k) | (0.99, 100) |
| | KL Loss | True |
| | KL Loss Coefficient | 0.2 |
| | Entropy Coefficient | 0 |
| | Dual Clip Loss | ture |
| | PPO Policy Clip | 0.2 |
| | Lambd | 0.95 |
| | Gamma | 1 |

**Generalization to multi-agent systems.**    For downstream evaluations, we assess all models on a comprehensive suite of mathematics and general QA benchmarks. In the single-agent setting, we utilize evaluation scripts from Qwen2.5-Math (Yang et al., 2024a;b) for MATH500, GSM8K, AQUA-RAT, AIME24, AMC23, and MMLU-STEM, while employing lm-evaluation-harness (Gao et al., 2024) for GPQA-Diamond. For multi-agent evaluations, we adopt the MASLab framework (Ye et al., 2025) across all benchmarks. To balance evaluation efficiency and accuracy, we set the maximum output length to 8,192 tokens for all settings.

## E  SCALING TO LARGER MODELS

To validate the scalability of the MARSHAL framework and ensure our findings are not specific to the 4B parameter scale, we extended our training to the larger Qwen3-8B model. We trained an 8B-parameter MARSHAL generalist agent using the exact same set of strategic games (*Tic-Tac-Toe*, *Kuhn Poker*, and *Mini Hanabi*) and hyperparameters as the 4B experiments.

**Strategic ability.**    First, we evaluate the model's proficiency on both training and held-out games. As shown in Table 6, the MARSHAL-8B agent achieves significant improvements over the Qwen3-8B base model across all environments. Notably, we observe the same strong generalization pattern seen in the 4B model: the agent generalizes effectively to held-out, complex games, achieving a drastic improvement in *Leduc Hold'em* (7.26% to 53.89%) and *Simple Hanabi* (4.55% to 37.27%).

Table 6: Strategic ability comparison on Qwen3-8B. The MARSHAL training yields consistent improvements on both training games and held-out testing games (*Connect Four*, *Leduc Hold'em*, *Simple Hanabi*).

| Model | Tic-Tac-Toe | Kuhn Poker | Mini Hanabi | Connect Four | Leduc Hold'em | Simple Hanabi |
|---|---|---|---|---|---|---|
| Qwen3-8B | 48.38 | 33.12 | 27.00 | 10.48 | 7.26 | 4.55 |
| MARSHAL (Generalist, 8B) | **54.05** | **44.49** | **55.28** | **21.55** | **53.89** | **37.27** |

**Generalization to multi-agent systems.**    We further evaluate whether these improved strategic capabilities translate to general reasoning benchmarks within multi-agent systems. We integrated

the MARSHAL-8B agent into both the competitive MAD framework and the cooperative AutoGen framework.

The results, detailed in Table 7, confirm the scalability of our approach. The MARSHAL-trained 8B model consistently outperforms the base model:

- In the competitive MAD setting, we observe an average improvement of 2.60%, with substantial gains on challenging math benchmarks like AIME (70.00% to 80.00%).
- In the cooperative AutoGen setting, the improvement is even more pronounced, with an average gain of 3.90%, and a significant boost on AIME (60.00% to 70.00%).

These results demonstrate that the MARSHAL algorithm and game selection scale stably to larger models, consistently unlocking cooperative and competitive reasoning capabilities.

Table 7: Generalization to multi-agent systems using Qwen3-8B. The MARSHAL generalist consistently improves performance on reasoning benchmarks in both competitive (MAD) and cooperative (AutoGen) settings.

| MAS | Model | Avg | MATH | GSM8K | AQUA | AIME | AMC | MMLU | GPQA |
|---|---|---|---|---|---|---|---|---|---|
| MAD | Qwen3-8B | 82.49 | 95.00 | 96.36 | 83.46 | 70.00 | 90.00 | 89.59 | 53.03 |
| | MARSHAL (Generalist, 8B) | **85.09** | **96.40** | **96.59** | 83.46 | **80.00** | **95.00** | **90.70** | **53.54** |
| AutoGen | Qwen3-8B | 79.68 | 88.80 | 95.91 | 83.07 | 60.00 | 89.19 | 89.30 | 51.52 |
| | MARSHAL (Generalist, 8B) | **83.58** | **94.40** | 95.00 | **85.04** | **70.00** | **95.00** | **90.04** | **55.56** |

## F    COMPARISON WITH SPIRAL: DECOUPLING ALGORITHM AND GAME ENVIRONMENTS

To rigorously decouple the contributions of our proposed learning algorithm and our selection of game environments, we conducted a set of controlled experiments comparing MARSHAL against SPIRAL baseline methods. We analyze two specific variations:

1. **Ablation on algorithm (RAE + our games):** We train an agent using SPIRAL's Role-conditioned Advantage Estimation (RAE) on our full selection of competitive and cooperative game environments. This isolates the impact of the advantage estimation technique.
2. **Ablation on games (MARSHAL algorithm + competitive games):** We train an agent using the MARSHAL algorithm but restricted to competitive-only games (*Tic-Tac-Toe* and *Kuhn Poker*), similar to SPIRAL. This isolates the impact of the cooperative games.

### F.1    STRATEGIC ABILITY COMPARISON

Table 8 presents the performance on both training and held-out games. We analyze the results through the lens of the two ablations:

**Ablation on algorithm.**    When replacing the MARSHAL algorithm with RAE (row 2), we observe a sharp performance decline in the cooperative games. Specifically, performance in *Mini Hanabi* falls from 54.33% to 24.93%, and in *Simple Hanabi* from 36.75% to 5.53%. This confirms that RAE struggles with the dense, multi-turn credit assignment required for cooperative settings, validating the necessity of MARSHAL's turn-level "sum-then-normalize" technique to unlock the potential of self-play learning.

**Ablation on games.**    When training only on competitive games (row 3), the agent naturally fails to acquire cooperative skills (low scores in *Hanabi*). This serves as a crucial control baseline for analyzing the downstream generalization results below.

### F.2    GENERALIZATION TO MULTI-AGENT SYSTEMS

We further evaluate how these variations affect generalization to standard reasoning benchmarks within multi-agent systems.

Table 8: Strategic ability comparison decoupling algorithm and game environments. MARSHAL (row 1) outperforms both the RAE-based agent (row 2) and the competitive-only agent (row 3), particularly in cooperative environments.

| Model | Tic-Tac-Toe | Kuhn Poker | Mini Hanabi | Connect Four | Leduc Hold'em | Simple Hanabi |
|---|---|---|---|---|---|---|
| MARSHAL (Full Method) | **53.13** | 40.05 | **54.33** | **19.98** | **54.56** | **36.75** |
| SPIRAL (RAE) + Our Games | 48.35 | 42.05 | 24.93 | 9.70 | 6.11 | 5.53 |
| MARSHAL Alg. + Adv. Games | 48.80 | **47.55** | 24.48 | 17.90 | 29.98 | 11.48 |

**Generalization to competitive systems (MAD).** As shown in Table 9, the full MARSHAL method achieves the highest average performance (75.96%). Notably, the RAE-based agent underperforms (74.23%), confirming the algorithmic superiority of MARSHAL. The competitive-only agent performs comparably to the full agent, which is expected as MAD is a competitive framework.

**Generalization to cooperative systems (AutoGen).** Table 10 reveals the critical role of the game environments. The agent trained on the competitive-only games suffers a significant performance drop in the cooperative AutoGen framework (Average 80.34% vs. 82.15% for MARSHAL). This confirms that cooperative games are not merely "more data"; they are a necessary component for generalizing skills to cooperative downstream tasks. Additionally, the RAE-based agent also underperforms, again solidifying the novelty of the algorithmic design of MARSHAL.

Table 9: Generalization to the competitive MAD framework.

| Model | Avg | MATH | GSM8K | AQUA | AIME | AMC | MMLU | GPQA |
|---|---|---|---|---|---|---|---|---|
| MARSHAL (Full Method) | **75.96** | **92.80** | 95.60 | 83.86 | **46.67** | **80.00** | **87.36** | 45.45 |
| SPIRAL (RAE) + Our Games | 74.23 | 92.40 | **96.06** | **84.25** | 36.67 | 77.50 | 86.79 | **45.96** |
| MARSHAL Alg. + Adv. Games | 75.24 | 92.20 | 95.60 | 83.46 | **46.67** | **80.00** | 86.88 | 41.92 |

Table 10: Generalization to the cooperative AutoGen framework. Note the drop in performance for the competitive-only games (row 3), highlighting the necessity of cooperative training.

| Model | Avg | MATH | GSM8K | AQUA | AIME | AMC | MMLU | GPQA |
|---|---|---|---|---|---|---|---|---|
| MARSHAL (Full Method) | **82.15** | **95.20** | 94.54 | **86.61** | **66.67** | **92.50** | 89.53 | **50.00** |
| SPIRAL (RAE) + Our Games | 80.39 | 94.80 | **94.77** | **86.61** | 60.00 | **92.50** | 88.57 | 45.45 |
| MARSHAL Alg. + Adv. Games | 80.34 | 94.20 | 94.24 | **86.61** | 60.00 | 90.00 | 89.37 | 47.98 |

**Conclusion.** These decoupling experiments confirms the superiority of MARSHAL's algorithmic design over SPIRAL's RAE, and that MARSHAL's successful generalization is not due to one factor alone. It requires both our selection of competitive and cooperative games to provide necessary learning signals, and our novel credit assignment algorithm to effectively learn from them.

# G COMPARISON WITH MT-GRPO

A concurrent work MT-GRPO (Zeng et al. (2025)) also identifies the limitations of trajectory-level estimation in the original GRPO and proposes a fine-grained turn-level strategy. While MT-GRPO shares our motivation, and is also a critical first step towards fine-grain credit assignment in multi-turn GRPO, our "sum-then-normalize" approach differs fundamentally from their strategy, offering distinct advantages in stability and scalability in the game tasks that we consider in this work.

## G.1 THEORETICAL DIFFERENCES

MT-GRPO relies on normalizing the turn-level rewards turn-by-turn (i.e., computing the mean and variance for the reward at step $t$ across the batch). We identify two structural limitations with this approach that MARSHAL addresses:

1. **The variable length problem:** In real-world reasoning and gaming tasks, trajectory lengths vary significantly. With turn-by-turn normalization, the effective batch size shrinks as shorter trajectories conclude. This causes the variance of the normalization statistics to increase for later turns in the sequence, introducing instability for training. In contrast, MARSHAL normalizes the cumulative return, which remains defined for the entire trajectory regardless of length.

2. **Immediate reward vs. long-term return:** MT-GRPO's local normalization focuses on the immediate step-wise reward relative to peers at that specific step. MARSHAL calculates the full Monte Carlo return first. By normalizing the cumulative outcome, we capture the long-term impact of actions relative to the global baseline, preserving critical long-term dependencies often obscured by local normalization.

## G.2 EMPIRICAL COMPARISON

To validate this theoretical analysis, we implemented MT-GRPO and compared it directly against MARSHAL. We trained a *Tic-Tac-Toe* specialist using both algorithms and evaluated their strategic proficiency across our full game suite.

**Strategic ability.** As shown in Table 11, the MT-GRPO agent exhibits notable performance drop across both the training game (*Tic-Tac-Toe*) and generalization to held-out games (e.g., *Connect Four*, *Hanabi*). In particular, the drop is significant in *Mini Hanabi* (50.48% to 36.67%), a game requiring consistent long-term coordination. This confirms that our "sum-then-normalize" formulation is more robust for the variable-length, multi-turn interactions inherent in strategic games.

**Generalization to MAS.** We further extended this comparison to the integration within multi-agent systems to assess if the stability issues in game training affect generalization. Table 12 present the results in the competitive MAD and cooperative AutoGen frameworks, respectively. MARSHAL consistently achieves higher average scores (75.01% vs. 73.77% in MAD; 80.15% vs. 79.10% in AutoGen) and outperforms MT-GRPO on key hard reasoning benchmarks like AIME and AMC. This indicates that the training proficiency provided by MARSHAL translates directly to better robust reasoning in general multi-agent tasks.

Table 11: Comparison between MARSHAL and MT-GRPO. The MARSHAL advantage estimation yields superior performance and generalization compared to the concurrent MT-GRPO method, validating the robustness of the "sum-then-normalize" approach.

| Model (Tic-Tac-Toe Specialist) | Tic-Tac-Toe | Kuhn Poker | Mini Hanabi | Connect Four | Leduc Hold'em | Simple Hanabi |
|---|---|---|---|---|---|---|
| MARSHAL (Ours) | **53.70** | 38.79 | **50.48** | **22.75** | **43.00** | **29.75** |
| MT-GRPO | 50.10 | **40.05** | 36.67 | 18.55 | 39.94 | 20.08 |

Table 12: Comparison of generalization to multi-agent systems between MARSHAL and MT-GRPO (using *Tic-Tac-Toe* specialists). MARSHAL achieves higher average performance and demonstrates greater stability in downstream tasks.

| MAS | Model (Tic-Tac-Toe Specialist) | Avg | MATH | GSM8K | AQUA | AIME | AMC | MMLU | GPQA |
|---|---|---|---|---|---|---|---|---|---|
| MAD | MARSHAL (Ours) | **75.01** | 92.20 | **96.06** | 83.07 | **43.33** | **82.50** | 86.76 | 41.12 |
| | MT-GRPO | 73.77 | **92.60** | 95.91 | **84.65** | 36.67 | 77.50 | **86.91** | **42.13** |
| AutoGen | MARSHAL (Ours) | **80.15** | 94.40 | **94.69** | **87.01** | **60.00** | **90.00** | **89.53** | **45.45** |
| | MT-GRPO | 79.10 | **95.40** | 94.62 | 85.04 | 56.67 | **90.00** | 89.02 | 42.93 |

## H HYPERPARAMETER ANALYSIS ON RESPONSE LENGTH PENALTY

To ensure efficient and stable reasoning within the context window limits, our reward design includes a response length penalty characterized by three hyperparameters: the minimum length $l_{min}$, the maximum threshold $l_{max}$, and the penalty weight $\alpha$.

**Selection of length constraints.** The length bounds were chosen based on practical engineering constraints:

- $l_{\min} = 11$: This is the hard lower bound determined by the minimum number of tokens required to generate a syntactically valid response structure with the correct format (e.g., `<think></think><action>...</action>`).
- $l_{\max} = 2048$: This soft upper bound is derived from the context window of the base model. Given Qwen3-4B's 32k context length, we allocated a budget of approximately 16 turns per game history ($32,768/16 = 2048$), which provides a sufficient horizon for the reasoning traces in our chosen game portfolio while preventing context overflow.

**Ablation on penalty weight $\alpha$.** To assess the sensitivity of the framework to the penalty weight $\alpha$, we conducted an ablation study training *Tic-Tac-Toe* specialists with $\alpha = 1.0$ (stronger regularization), $\alpha = 0.5$ (default), and $\alpha = 0.0$ (no regularization). The results are detailed in Table 13.

Table 13: Ablation study on the length penalty weight $\alpha$. The default setting ($\alpha = 0.5$) strikes a balance between performance and conciseness. Removing the penalty ($\alpha = 0$) leads to overly verbose responses and performance degradation in cooperative games like *Hanabi*. "OL" denotes the percentage of games lost due to overlong responses.

| Model Setup | Strategic Performance | | | | | | Response Statistics | | |
|---|---|---|---|---|---|---|---|---|---|
| | Tic-Tac-Toe | Kuhn Poker | Mini Hanabi | Connect Four | Leduc Hold'em | Simple Hanabi | Avg Len | OL in Mini Hanabi | OL in Simple Hanabi |
| $\alpha = 0.5$ (Default) | 53.70 | 38.79 | **50.48** | 22.75 | 43.00 | **29.75** | 1700 | 9.6% | 14.3% |
| $\alpha = 1.0$ (Strong) | **54.90** | **42.82** | 47.53 | 18.25 | **50.46** | 26.05 | 1657 | **7.7%** | **13.9%** |
| $\alpha = 0.0$ (None) | 54.65 | 40.95 | 38.18 | **23.35** | 28.61 | 20.10 | 1954 | 20.4% | 27.3% |

**Analysis.** The results yield two key conclusions:

1. **Robustness:** The framework is robust to reasonable variations in $\alpha$. Increasing the regularization to $\alpha = 1.0$ maintains comparable strategic performance to the default setting, indicating that the method is not overly sensitive to the precise magnitude of the penalty.
2. **Necessity of regularization:** The length penalty is crucial for stability. When removed ($\alpha = 0$), the average response length increases significantly (from 1700 to 1954 tokens). This verbosity leads to a sharp performance drop in cooperative, theory-of-mind-demanding games like *Mini Hanabi* (50.48% to 38.18%). This degradation correlates directly with a dramatic increase in the "Overlong Response" failure rate (9.6% to 20.4%), confirming that the penalty effectively prevents the model from degenerating into computationally wasteful loops.

## I  GAME OBSERVATION AND PROMPT

***Tic-Tac-Toe*** For *Tic-Tac-Toe*, we provide the agent with a complete observation of the 3x3 game board. The state of each cell—whether it is empty, occupied by 'X', or occupied by 'O'—is explicitly provided. The prompt clearly indicates which player's turn it is ('X' or 'O') and presents the current board state, asking the agent to select coordinates for its next move from the available empty cells. For example, the game begins with a prompt that provides the empty 3x3 grid and asks the agent to make the first move (Listing 1).

Listing 1: Prompt for *Tic-Tac-Toe*.

```
system_prompt:
You are an AI agent that makes optimal decisions to win in the game of
Tic-Tac-Toe.

user_prompt:
GAME RULES:
1. Tic-Tac-Toe is a two-player board game played on a three-by-three grid
. The grid is 0-indexed, where (0,0) is the top-left corner and (2,2) is
the bottom-right corner.
```

```
2. Two players take turns placing their marks X and O in empty cells of
the grid.
3. The player who first places three of their marks in a horizontal,
vertical, or diagonal line wins.
4. If all cells are filled and no player wins, the game ends in a draw.

PLAYER INFORMATION:
1. Your mark is X. You are competing with another player controlling the
mark O.
2. In each of your turns:
   a. The game state demonstrates the current board with a three-line
   text grid, where 'X' and 'O' are the marks of the two players, and '_'
    represents empty cells.
   b. You need to chose an action to place your mark in an empty cell,
   based on the given game state and the history of your decisions.
   c. All legal actions for the current turn are provided in the format
   of '<X({row},{column})>', where 'X' is your mark, and {row} and {
   column} are integers indicating the row and column of the cell to
   place your mark.

RESPONSE INSTRUCTIONS:
Always choose only one action from the legal actions and output '<answer
>{your chosen action}</answer>' with no extra text after you finish the
thinking process. For example, '<answer><X(0,0)></answer>'. Strictly
follow the above format and keep your thinking process concise. Responses
 that do not follow the format will result in immediate loss of the game.

Information of Turn-1:
This is your turn. The game state and legal actions for this turn are
provided below. Please choose your action and strictly follow the given
output format in the response instructions.

GAME STATE:
___
___
___

LEGAL ACTIONS:
<X(0,0)>, <X(0,1)>, <X(0,2)>, <X(1,0)>, <X(1,1)>, <X(1,2)>, <X(2,0)>, <X
(2,1)>, <X(2,2)>.
```

*Kuhn Poker*   For *Kuhn Poker*, we provide the agent with the observation consists of its single
private card (e.g., Jack, Queen, or King) and the complete history actions of two players. The prompt
is structured to provide this context clearly, asking the agent to decide on its next action (pass or
bet) based on its private information and the betting history. For example, the initial prompt at the
beginning of game provides the agent's private card and the empty action history, asking for the first
move (Listing 2).

Listing 2: Prompt for *Kuhn Poker*.

```
system_prompt:
You are an AI agent that makes optimal decisions to win in the game of
Kuhn Poker.

user_prompt:
GAME RULES:
1. Kuhn Poker is a two-player card game. The deck includes only three
cards: King (K) > Queen (Q) > Jack (J).
2. At the start of each game, both player_0 and player_1 place 1 chip
into the pot as a blind ante.
3. Each player is dealt a private card, and the third card is set aside
unseen.
4. The two players take turns acting, starting with player_0. A player
can choose to:
```

```
      a. <PASS>: place no additional chips into the pot.
      b. <BET>: place 1 additional chip into the pot.
5. If a player chooses to <PASS> after the other player's <BET>, the
betting player wins the pot.
6. If both players choose to <PASS> or both players choose to <BET>, the
player with the higher card wins the pot.

PLAYER INFORMATION:
1. You are player_0. You are competing with player_1.
2. In each of your turns:
   a. The game state shows your private card and the betting history.
   b. You need to choose an action based on your card and the current
   game state.
   c. All legal actions for the current turn are provided in the format
   of '<PASS>' or '<BET>'.

RESPONSE INSTRUCTIONS:
Always choose only one action from the legal actions and output '<answer
>{your chosen action}</answer>' with no extra text after you finish the
thinking process. For example, '<answer><PASS></answer>'. Strictly follow
 the above format and keep your thinking process concise. Responses that
do not follow the format will result in immediate loss of the game.

Information of Turn-1:
This is your turn. The game state and legal actions for this turn are
provided below. Please choose your action and strictly follow the given
output format in the response instructions.

GAME STATE:
1. Blind ante: both player_0 and player_1 place 1 chip into the pot.
2. Deal: your card is Jack (J).

LEGAL ACTIONS:
<PASS>, <BET>.
```

*Hanabi*    For the *Hanabi* variants, a cooperative imperfect information game, we provide the agent
with a unique observation. An agent observes the cards held by all other players but remains unaware
of its own hand. The observation also includes the number of remaining information and life tokens,
the cards successfully played on the board (fireworks), and the contents of the discard pile. The
prompt is highly structured, detailing the game rules, player-specific information, and strict response
formatting. For training, we use *Mini Hanabi*, a simplified 2-player version with 2 colors, 2 ranks, 3-
card hands, and 3 information/life tokens. For out-of-distribution evaluation, we use a more complex
variant, *Simple Hanabi*, which features 3 colors, 2 ranks, 5-card hands, 8 information tokens and 3
life tokens. For example, an initial prompt details the starting setup of the chosen variant, including
all visible cards and token counts, before asking the first player to make the first move (Listing 3).

Listing 3: Prompt for *Mini Hanabi*.

```
system_prompt:
You are an AI agent that makes optimal decisions to achieve the highest
score in the game of Hanabi.

user_prompt:
GAME RULES:
1. Hanabi is a cooperative card game for 2 players, player 0 and player
1.
2. The deck consists of 2 colors: Red(denoted by R), Yellow(denoted by Y)
, with ranks ranging from 1 to 2. Each color contains 4 cards: three of
rank 1, and one of rank 2, for a total of 8 cards.
3. Each player holds 3 cards in hand. Players can observe the hand of the
 other player, but not their own.
4. There are 3 information tokens and 3 life tokens shared by both
players.
```

5. The objective is to play cards in ascending order of rank, from 1 to
2, to their corresponding color stacks, hence achieving the 'Fireworks'.
6. The players take turns to take one of the following actions:
    a. <Play 'i'>: play the i-th card from the player's own hand (0-
    indexed). If the card is sequential to the top card of its
    corresponding color stack, the move is valid and the card is added to
     the top of the stack, then both players receive 1 point. Otherwise,
    a life token is lost.
    b. <Discard 'i'>: discard the i-th card from the player's own hand
    and gain one information token.
    c. <Reveal player +1 color 'c'>: spend one information token to
    reveal all cards of color 'c' in the other player's hand.
    d. <Reveal player +1 rank 'r'>: spend one information token to reveal
     all cards of rank 'r' in the other player's hand.
7. After playing or discarding, the player receives a new card from the
deck (if remaining).
8. The game ends when:
    a. If all color stacks are completed (i.e., all cards of rank 2 are
    played to their corresponding color stacks), then both players finish
     the game with the highest possible total score of 4.
    b. If deck is depleted, both players finish the game with a total
    score which equals the sum of the highest ranks of each color stack.
    c. If all life tokens are lost before the above two conditions are
    met, then both players lose all points they have earned so far, and
    finish the game with a total score of 0.

PLAYER INFORMATION:
1. You will be playing as the player 0.
2. In each of your turns, you will be provided with the current game
state information, including the remaining life tokens and information
tokens, the current color stacks, the remaining deck size, the discard
pile, the hand of the other player, and the revealed information on your
own hand.
3. Known cards are denoted by their color and rank. For example, 'R2'
means a red card of rank 2. 4. The current color stacks are represented
by the top card of each color stack. In particular, rank 0 denotes an
empty stack. For example, 'Y0' means the yellow stack is still empty.

RESPONSE INSTRUCTIONS:
Always choose only one action from the legal actions and output '<answer
>{your chosen action}</answer>' with no extra text after you finish the
thinking process. For example, '<answer><Discard Card 0></answer>'.
Strictly follow the above format and keep your thinking process concise.
Responses that do not follow the format will result in immediate loss of
all life tokens and end of the game.

Information of Turn-1:
This is your turn. The game state and legal actions for this turn are
provided below. Please choose your action and strictly follow the given
output format in the response instructions.

GAME STATE:
1. There are 3 life tokens and 3 information tokens remaining.
2. The top of the color stacks are: R0 Y0.
3. 2 cards remain in the draw pile.
4. The discard pile currently contains: None.
5. The other player's hand:
    - Card 0 (Y1): the other player believes it is one of the colors [R,
    Y] and one of the ranks [1, 2].
    - Card 1 (R1): the other player believes it is one of the colors [R,
    Y] and one of the ranks [1, 2].
    - Card 2 (R1): the other player believes it is one of the colors [R,
    Y] and one of the ranks [1, 2].
6. Your own hand, based on the revealed information:
    - Card 0: one of the colors [R, Y] and one of the ranks [1, 2].

```
      - Card 1: one of the colors [R, Y] and one of the ranks [1, 2].
      - Card 2: one of the colors [R, Y] and one of the ranks [1, 2].

LEGAL ACTIONS:
<Play card 0>, <Play card 1>, <Play card 2>, <Reveal player +1 color R>,
<Reveal player +1 color Y>, <Reveal player +1 rank 1>.
```

*Connect Four*   For *Connect Four*, we provide the agent with a complete observation of the 6x7 game board, showing the positions of all 'X' and 'O' marks. The prompt is designed to be comprehensive, initially presenting the rules of the game (connecting four pieces to win, draw conditions). It then informs the agent of its assigned mark ('X' or 'O') and the opponent's mark. For each turn, the prompt presents the current board state and asks the agent to choose a column (0-6) to drop its piece. For example, the game begins with a prompt that provides the empty 6x7 grid, explains the rules and player marks, and asks the first player to choose a column (Listing 4).

Listing 4: Prompt for *Connect Four*.

```
system_prompt:
You are an AI agent that makes optimal decisions to win in the game of
Connect Four.

user_prompt:
GAME RULES:
1. Connect Four is a two-player board game played on a 6x7 grid. Players
take turns dropping their pieces into columns.
2. The goal is to connect four of your pieces horizontally, vertically,
or diagonally.
3. Pieces fall to the bottom of the column or stack on top of existing
pieces.
4. The first player to connect four pieces wins. If the board fills up
without a winner, it's a draw.

PLAYER INFORMATION:
1. Your mark is X. You are competing with another player controlling the
mark O.
2. In each of your turns:
   a. The game state shows the current board as a 6x7 grid.
   b. You need to choose a column (0-6) to drop your piece, where 0
   denotes the leftmost column, 6 denotes the rightmost column.
   c. All legal actions are provided as '<X({column})>', where 'X' is
   your mark, and {column} is the column number.

RESPONSE INSTRUCTIONS:
Always choose only one action from the legal actions and output '<answer
>{your chosen column}</answer>' with no extra text after you finish the
thinking process. For example, '<answer><X(3)></answer>'. Strictly follow
 the above format and keep your thinking process concise. Responses that
do not follow the format will result in immediate loss of the game.

Information of Turn-1:
This is your turn. The game state and legal actions for this turn are
provided below. Please choose your action and strictly follow the given
output format in the response instructions.

GAME STATE:
-------
-------
-------
-------
-------
-------

LEGAL ACTIONS:
```

```
<X(0)>, <X(1)>, <X(2)>, <X(3)>, <X(4)>, <X(5)>, <X(6)>.
```

**Leduc Hold'em**   For *Leduc Hold'em*, we provide the agent with the observation which contains all necessary information for decision-making. The observation is more complex due to two betting rounds and a public card. In the first round, the agent observes its private card and the betting history. In the second round, after a public card is revealed, the observation is updated to include this board card. The prompt provides the agent with its private card, the current betting history, and the public card if revealed, requiring it to make an action in the context of the evolving game state. For example, the initial prompt at the beginning of game provides the agent with its private card and asks for an action in the first betting round before any actions have been taken (Listing 5).

Listing 5: Prompt for *Leduc Hold'em*.

```
system_prompt:
You are an AI agent that makes optimal decisions to win in the game of
Leduc Poker.

user_prompt:
GAME RULES:
1. Leduc poker is a two-player card game. The deck includes only six
cards: two pairs of King (K), Queen (Q), and Jack (J).
2. At the start of each game, both player_0 and player_1 place 1 chip
into the pot as a blind ante.
3. Each player is dealt one private card from the deck, and the remaining
 cards are set aside unseen.
4. The game has two betting rounds. When the first round ends, one public
 card from the remaining cards of the deck is revealed to both players.
5. The two players take turns acting in the betting rounds, both starting
 with player_0. A player can choose to:
    a. <FOLD>: stop betting and the other player wins the pot.
    b. <CALL>: match the current bet. If no bet has been made in the
    current round, this is equivalent to checking.
    c. <RAISE>: first match the current bet and then add 'n' chips to the
     bet, where 'n=2' in the first round and 'n=4' in the second round.
    If no bet has been made in the current round, this is equivalent to
    betting 'n' chips.
6. A maximum of two <RAISE>s are allowed in each round. Each round ends
when both players have acted and their bets are equal.
7. If a player chooses to <FOLD>, the other player wins the pot.
8. If neither player chooses to <FOLD>, the second round ends with a
showdown:
    a. If a player has a pair (private card = public card), the player
    wins the pot.
    b. If neither player has a pair, the player with the higher card (K >
     Q > J) wins the pot.
    c. If two players have the same card, the players split the pot.

PLAYER INFORMATION:
1. You are player_0. You are competing with player_1.
2. In each of your turns:
   a. The game state shows your private card, public card (if revealed),
   and the betting history.
   b. You need to choose an action based on your cards and the current
   game state.
   c. All legal actions for the current turn are provided in the format
   of '<FOLD>', '<CALL>', or '<RAISE>'.

RESPONSE INSTRUCTIONS:
Always choose only one action from the legal actions and output '<answer
>{your chosen action}</answer>' with no extra text after you finish the
thinking process. For example, '<answer><CALL></answer>'. Strictly follow
 the above format and keep your thinking process concise. Responses that
do not follow the format will result in immediate loss of the game.
```

```
Information of Turn -1:
This is your turn. The game state and legal actions for this turn are
provided below. Please choose your action and strictly follow the given
output format in the response instructions.

GAME STATE:
1. Blind ante: both player_0 and player_1 place 1 chip into the pot.
2. Deal: your card is J.

LEGAL ACTIONS:
<CALL >, <RAISE >.
```

## J  FULL RESULTS

In this section, we present the raw (unnormalized) results for all experiments in the main text.

Table 14: Full results of game-play return in training games

| Model | Tic-Tac-Toe | | Kuhn Poker | Mini Hanabi |
| | MCTS (100 Simulations) | MCTS (1000 Simulations) | | |
|---|---|---|---|---|
| Qwen3-4B | 0.403/-0.629 | -0.374/-0.687 | -0.148/-0.141 | 1.200 |
| SPIRAL | 0.470/-0.624 | -0.297/-0.717 | -0.301/-0.149 | 1.494 |
| MARSHAL | | | | |
|   Tic-Tac-Toe | 0.506/-0.358 | -0.278/-0.379 | -0.119/-0.142 | 2.019 |
|   Kuhn Poker | 0.397/-0.490 | -0.388/-0.538 | -0.107/-0.103 | 1.799 |
|   Mini Hanabi | 0.438/-0.644 | -0.350/-0.702 | -0.123/-0.130 | 2.222 |
|   Generalist | 0.544/-0.419 | -0.302/-0.450 | -0.114/-0.141 | 2.173 |
| MARSHAL (*w/ Fixed Opponent*) | | | | |
|   Tic-Tac-Toe | 0.760/-0.161 | -0.378/-0.162 | -0.146/-0.090 | 1.397 |
|   Kuhn Poker | -1.000/-1.000 | -1.000/-1.000 | -0.114/-0.117 | 0.000 |
|   Mini Hanabi | - | - | - | - |
| MARSHAL (*w/o Turn-Level Advantage Estimator*) | | | | |
|   Tic-Tac-Toe | 0.492/-0.517 | -0.292/-0.567 | -0.104/-0.091 | 1.392 |
|   Kuhn Poker | 0.267/-0.607 | -0.426/-0.625 | -0.074/-0.106 | 1.666 |
|   Mini Hanabi | 0.351/-0.788 | -0.400/-0.811 | -0.133/-0.084 | 2.128 |
| MARSHAL (*w/o Agent-Specific Advantage Normalization*) | | | | |
|   Tic-Tac-Toe | 0.654/-0.376 | -0.213/-0.366 | -0.127/-0.126 | 1.764 |
|   Kuhn Poker | 0.391/-0.509 | -0.446/-0.567 | -0.116/-0.109 | 1.607 |
|   Mini Hanabi | 0.363/-0.732 | -0.364/-0.764 | -0.119/-0.128 | 2.100 |

Table 15: Full results on game-play return in testing games

| Model | Connect Four | | Leduc Hold'em | Simple Hanabi |
| | MCTS (10 Simulations) | MCTS (100 Simulations) | | |
|---|---|---|---|---|
| Qwen3-4B | -0.462/-0.912 | -0.902/-0.998 | -0.629/-0.691 | 0.833 |
| SPIRAL | -0.383/-0.833 | -0.868/-0.995 | -0.870/-0.706 | 0.836 |
| MARSHAL | | | | |
|   Tic-Tac-Toe | -0.387/-0.703 | -0.858/-0.974 | -0.518/-0.702 | 1.785 |
|   Kuhn Poker | -0.448/-0.746 | -0.874/-0.995 | -0.460/-0.327 | 1.761 |
|   Mini Hanabi | -0.465/-0.885 | -0.880/-0.998 | -0.736/-0.403 | 2.036 |
|   Generalist | -0.421/-0.780 | -0.871/-0.979 | -0.483/-0.487 | 2.205 |
| MARSHAL (*w/ Fixed Opponent*) | | | | |
|   Tic-Tac-Toe | -0.593/-0.887 | -0.939/-0.990 | -0.632/-0.616 | 0.733 |
|   Kuhn Poker | -1.000/-1.000 | -1.000/-1.000 | -1.124/-1.003 | 0.000 |
|   Mini Hanabi | - | - | - | - |
| MARSHAL (*w/o Turn-Level Advantage Estimator*) | | | | |
|   Tic-Tac-Toe | -0.465/-0.754 | -0.843/-0.989 | -0.622/-0.553 | 1.143 |
|   Kuhn Poker | -0.408/-0.783 | -0.899/-0.998 | -0.789/-0.663 | 1.379 |
|   Mini Hanabi | -0.495/-0.939 | -0.886/-0.998 | -0.555/-0.483 | 1.841 |
| MARSHAL (*w/o Agent-Specific Advantage Normalization*) | | | | |
|   Tic-Tac-Toe | -0.492/-0.790 | -0.882/-0.983 | -0.594/-0.460 | 1.303 |
|   Kuhn Poker | -0.460/-0.790 | -0.908/-0.992 | -0.753/-0.769 | 1.345 |
|   Mini Hanabi | -0.358/-0.898 | -0.902/-0.998 | -0.664/-0.389 | 1.925 |

Table 16: Full results on general benchmarks in standard single agent setting

| Model | MATH | GSM8K | AQUA | AIME | AMC | MMLU | GPQA |
|---|---|---|---|---|---|---|---|
| Qwen3-4B | 87.60 | 94.60 | 39.80 | 36.70 | 70.00 | 57.10 | 39.39 |
| SPIRAL | 87.50 | 94.80 | 51.20 | 36.70 | 80.00 | 58.70 | 37.37 |
| MARSHAL | | | | | | | |
|   Tic-Tac-Toe | 89.10 | 95.20 | 46.50 | 40.00 | 77.50 | 57.60 | 38.89 |
|   Kuhn Poker | 87.80 | 94.50 | 48.40 | 33.30 | 72.50 | 59.30 | 33.84 |
|   Mini Hanabi | 88.10 | 94.70 | 48.00 | 43.30 | 65.00 | 58.90 | 36.36 |
|   Generalist | 89.90 | 94.60 | 52.00 | 33.30 | 75.00 | 59.90 | 34.85 |
| MARSHAL (*w/ Fixed Opponent*) | | | | | | | |
|   Tic-Tac-Toe | 89.10 | 94.80 | 53.90 | 50.00 | 72.50 | 63.00 | 37.88 |
|   Kuhn Poker | 86.20 | 94.00 | 45.70 | 33.30 | 72.50 | 53.80 | 33.84 |
|   Mini Hanabi | - | - | - | - | - | - | - |
| MARSHAL (*w/o Turn-Level Advantage Estimator*) | | | | | | | |
|   Tic-Tac-Toe | 88.80 | 94.80 | 48.80 | 40.00 | 72.50 | 57.70 | 36.36 |
|   Kuhn Poker | 88.60 | 94.00 | 48.40 | 40.00 | 72.50 | 57.90 | 34.34 |
|   Mini Hanabi | 87.80 | 94.50 | 48.40 | 36.70 | 70.00 | 59.30 | 36.87 |
| MARSHAL (*w/o Agent-Specific Advantage Normalization*) | | | | | | | |
|   Tic-Tac-Toe | 88.20 | 94.50 | 53.10 | 36.70 | 77.50 | 58.20 | 39.39 |
|   Kuhn Poker | 88.10 | 94.80 | 53.90 | 40.00 | 75.00 | 58.70 | 35.86 |
|   Mini Hanabi | 88.60 | 93.90 | 46.10 | 40.00 | 75.00 | 57.00 | 38.89 |

Table 17: Full results on general benchmarks using the MAD framework

| Model | MATH | GSM8K | AQUA | AIME | AMC | MMLU | GPQA |
|---|---|---|---|---|---|---|---|
| Qwen3-4B | 90.20 | 95.91 | 80.71 | 40.00 | 75.00 | 87.42 | 37.88 |
| SPIRAL | 91.60 | 95.45 | 81.89 | 40.00 | 77.50 | 87.01 | 40.40 |
| MARSHAL | | | | | | | |
|   Tic-Tac-Toe | 92.20 | 96.06 | 83.07 | 43.33 | 82.50 | 86.76 | 41.12 |
|   Kuhn Poker | 91.60 | 96.21 | 82.68 | 40.00 | 82.50 | 87.39 | 41.41 |
|   Mini Hanabi | 91.40 | 95.60 | 82.68 | 43.33 | 77.50 | 87.04 | 38.38 |
|   Generalist | 92.80 | 95.60 | 83.86 | 46.67 | 80.00 | 87.36 | 45.45 |
| MARSHAL (*w/ Fixed Opponent*) | | | | | | | |
|   Tic-Tac-Toe | 94.80 | 94.54 | 85.04 | 53.33 | 90.00 | 88.57 | 42.93 |
|   Kuhn Poker | 95.00 | 94.24 | 86.22 | 53.33 | 85.00 | 88.54 | 45.45 |
|   Mini Hanabi | - | - | - | - | - | - | - |
| MARSHAL (*w/o Turn-Level Advantage Estimator*) | | | | | | | |
|   Tic-Tac-Toe | 93.80 | 94.77 | 85.04 | 60.00 | 85.00 | 88.98 | 47.98 |
|   Kuhn Poker | 95.00 | 94.54 | 86.22 | 56.67 | 90.00 | 89.05 | 50.00 |
|   Mini Hanabi | 94.20 | 94.62 | 85.43 | 53.33 | 90.00 | 88.51 | 46.97 |
| MARSHAL (*w/o Agent-Specific Advantage Normalization*) | | | | | | | |
|   Tic-Tac-Toe | 94.40 | 94.62 | 84.65 | 56.67 | 87.50 | 89.33 | 46.97 |
|   Kuhn Poker | 95.00 | 94.69 | 84.25 | 56.67 | 85.00 | 89.62 | 47.47 |
|   Mini Hanabi | 94.20 | 94.47 | 87.01 | 63.33 | 90.00 | 88.92 | 43.94 |

Table 18: Full results on general benchmarks using the AutoGen framework

| Model | MATH | GSM8K | AQUA | AIME | AMC | MMLU | GPQA |
|---|---|---|---|---|---|---|---|
| Qwen3-4B | 93.40 | 94.69 | 85.04 | 56.67 | 87.50 | 89.21 | 47.47 |
| SPIRAL | 94.20 | 94.47 | 86.61 | 60.00 | 87.50 | 91.60 | 45.96 |
| MARSHAL | | | | | | | |
|   Tic-Tac-Toe | 94.40 | 94.69 | 87.01 | 60.00 | 90.00 | 89.53 | 45.45 |
|   Kuhn Poker | 95.80 | 94.39 | 86.61 | 63.33 | 92.50 | 89.65 | 48.48 |
|   Mini Hanabi | 94.40 | 94.54 | 86.22 | 66.67 | 95.00 | 88.98 | 44.95 |
|   Generalist | 95.20 | 94.54 | 86.61 | 66.67 | 92.50 | 89.53 | 50.00 |
| MARSHAL (*w/ Fixed Opponent*) | | | | | | | |
|   Tic-Tac-Toe | 92.40 | 95.38 | 83.07 | 43.33 | 80.00 | 87.11 | 43.94 |
|   Kuhn Poker | 90.20 | 95.83 | 83.46 | 33.33 | 72.50 | 86.69 | 42.42 |
|   Mini Hanabi | - | - | - | - | - | - | - |
| MARSHAL (*w/o Turn-Level Advantage Estimator*) | | | | | | | |
|   Tic-Tac-Toe | 91.00 | 95.75 | 82.28 | 40.00 | 72.50 | 86.34 | 42.93 |
|   Kuhn Poker | 91.80 | 95.45 | 82.68 | 36.67 | 82.50 | 86.79 | 41.92 |
|   Mini Hanabi | 91.20 | 95.91 | 83.86 | 40.00 | 75.00 | 87.33 | 41.92 |
| MARSHAL (*w/o Agent-Specific Advantage Normalization*) | | | | | | | |
|   Tic-Tac-Toe | 91.20 | 95.83 | 83.46 | 40.00 | 80.00 | 87.14 | 38.58 |
|   Kuhn Poker | 91.20 | 95.53 | 84.65 | 36.67 | 75.00 | 86.63 | 43.94 |
|   Mini Hanabi | 92.60 | 96.06 | 82.68 | 33.33 | 80.00 | 87.11 | 43.43 |

