# OpenReview forum: "MARSHAL: Incentivizing Multi-Agent Reasoning via Self-Play with Strategic LLMs"
_ICLR.cc/2026/Conference — ICLR 2026 Poster_

### Official Review · Reviewer_YdbT · 2025-10-31

**Soundness:** 3
**Presentation:** 3
**Contribution:** 3
**Rating:** 8
**Confidence:** 4

**Summary:**

This paper introduces MARS (Multi-agent RL method for LLMs), a new framework for training Large Language Models in multi-agent environments. The method includes:
1) the introduction of turn-level rewards to enable more fine-grained credit assignment,
2) an agent role-based advantage normalization technique, which aims to reduce training variance by calibrating advantage signals relative to the performance of rollouts from the same role,
3) self-play against a variety of partners opponents in competitive games.

The authors conduct experiments using Qwen3-4B as the backbone model. They train specialist agents on individual games representing diverse dynamics: Tic-Tac-Toe (competitive, deterministic), Kuhn Poker (competitive, uncertain), and Mini Hanabi (cooperative, uncertain). They also train a generalist agent (MARS-multi) on all games simultaneously. Evaluation is performed on held-out games with similar dynamics (Connect Four, Leduc Hold'em, Simple Hanabi). The results indicate that MARS specialist agents outperform baselines (SPIRAL, Qwen3-4B) on their respective held-out tasks. The paper also evaluates the agents on general STEM reasoning benchmarks, finding mixed results in a single-agent setting but small, consistent improvements when integrated into a multi-agent framework (AutoGen). The ablation suggests playing with mixed opponents is critical.

**Strengths:**

- The algorithmic contributions are very clear and well motivated.  Turn-level reward shaping for credit assignment and role-conditioned advantage normalization are simple, intuitive modifications to GRPO that target known pain points (variance, difficulty credit assignment).
- The experimental design is solid, as the choice of training and testing environments is commendable. The authors systematically cover competitive, cooperative, deterministic, and uncertain dynamics. Evaluating on held-out games (e.g., training on Tic-Tac-Toe and testing on Connect Four) provides a solid test of generalization to similar dynamics, which is a key claim of the paper.
- The results are positive when tested in case of generalization. We see that MARS agents outperform baselines on held-out games (Connect Four, Leduc Hold'em) is a significant result. This suggests the method is not just overfitting to the training games but is learning underlying strategies for that specific dynamics.

**Weaknesses:**

- All experiments use small, two-player environments with constrained action/communication spaces. This limits external validity; N-player settings introduce additional non-stationarity, credit assignment, and population-diversity challenges, so evidence scaling beyond 2-player cases is important to substantiate generality claims.

**Questions:**

- Ablation results in Table 5 shows that without either turn-level reward or agent-specific advantage normalization resulted in significant performance drop in Tic/Kuhn trained models, but not necessarily true for Hanabi-trained model in Kuhn Poker and Leduc Hold'em settings, do you have any intuition on why that is the case?

---

> ### Author Response · Authors · 2025-11-21
> **Response to Reviewer YdbT**
>
> We would like to express our sincere appreciation for the reviewer's thorough and detailed feedback. We have carefully studied each point and believe the resulting revisions have substantially improved the manuscript. The modifications to our revised manuscript are highlighted in blue. We address the specific points raised by the reviewer below.
>
> ---
>
> > W1: All experiments use small, two-player environments with constrained action/communication spaces. This limits external validity; N-player settings introduce additional non-stationarity, credit assignment, and population-diversity challenges, so evidence scaling beyond 2-player cases is important to substantiate generality claims.
>
> We thank the reviewer for this thoughtful comment regarding the scope of our game environments. We agree that N-player settings introduce higher degrees of non-stationarity and complexity, and scaling to such environments is an important next step for the field.
>
> However, we respectfully suggest that the use of constrained 2-player environments actually highlights the efficiency and power of our approach. Our central finding is that a diverse set of simple games (e.g., Tic-Tac-Toe, Hanabi) is sufficient to incentivize generalizable multi-agent skills.
>
> The strongest evidence for our external validity lies in our generalization results in multi-agent system (Table 1 in our revised paper). Despite training in these "constrained" 2-player games, our agents demonstrated notable performance gains when transferred to MASs on general reasoning tasks.
>
> This suggests that the fundamental cognitive patterns of cooperation and competition can be learned in simpler settings and successfully scaled to complex ones. We have updated the Discussion section to explicitly acknowledge N-player scaling as a key frontier for future work, while emphasizing that 2-player prototypes serve as a highly efficient foundational curriculum.
>
> > Q1: Ablation results in Table 5 shows that without either turn-level reward or agent-specific advantage normalization resulted in significant performance drop in Tic/Kuhn trained models, but not necessarily true for Hanabi-trained model in Kuhn Poker and Leduc Hold'em settings, do you have any intuition on why that is the case?
>
> We thank the reviewer for this insightful question regarding the ablation dynamics. We attribute the Hanabi-trained model's relative robustness to two intrinsic properties of the game that differ from the adversarial settings.
>
> In the cooperative Hanabi game, rewards are shared globally. Whenever a valid card is played, both players receive the same +1 reward. Consequently, the return distributions for different players are statistically similar. Therefore, removing agent-specific normalization (i.e., merging the player baselines) introduces minimal influence.
>
> However, in Tic-Tac-Toe, the first-move agent has an inherent advantage, leading to a different return distribution from the second-move player. Similarly, the second-move player in Kuhn poker has an inherent advantage and results in different return distribution. The differences in return distribution necessitate agent-specific normalization and lead to performance improvement.
>
> To substantiate this, we have added a histogram analysis in Section 4.5 comparing the return distributions of different players. The plots clearly show that returns are highly distinct in Tic-Tac-Toe but overlapping in Hanabi, perfectly explaining why the ablation effect is smaller in Hanabi.
>
> ---
>
> We thank the reviewer again for their careful and detailed review, which has undoubtedly improved the quality of our paper. We believe that our responses and the corresponding revisions have clarified the points raised and strengthened the overall contribution. We express our sincere gratitude for the reviewer's support and appreciation of our contributions.

---

### Official Review · Reviewer_dt93 · 2025-11-01

**Soundness:** 2
**Presentation:** 3
**Contribution:** 2
**Rating:** 6
**Confidence:** 3

**Summary:**

This paper introduces MARS, an RL framework for training LLMs via self-play using a mixed curriculum of cooperative (Hanabi) and competitive (Tic-Tac-Toe, Kuhn Poker) games.1 The authors claim this training, supported by two methodological tweaks to the GRPO algorithm ("turn-level advantage estimation" and "agent-specific advantage normalization") 1, leads to skills that generalize. The experimental results, particularly the transfer of these skills to downstream multi-agent systems like AutoGen and MAD 1, are quite strong and genuinely interesting.

However, I have some *very significant* concerns about the paper's framing, its engagement with closely related prior work, and the experimental design, all of which muddy the central claims of methodological novelty.

**Strengths:**

This is a solid piece of work, and I want to start by highlighting what I think is really excellent here.

1. **Originality & Significance (The Core Idea):** The most significant contribution here, in my view, is the demonstration of *skill transfer from a simple game curriculum to complex, downstream multi-agent systems*. This is a really big deal. Prior work like SPAG or SPIRAL showed self-play can boost *single-agent* reasoning benchmarks, but this paper is the first I've seen to convincingly argue that these skills transfer to *cooperative and competitive software systems* like AutoGen and MAD. The 10.0% gain on AIME and 12.5% on GPQA-Diamond are non-trivial and suggest this is a very promising research direction.
2. **Quality (Experimental Rigor):** The experimental setup is high-quality and, in one specific area, exceptionally well-designed.
   - The evaluation is comprehensive: held-out games (Table 1) , single-agent benchmarks (Table 2) , and downstream MAS (Table 2).
   - **Skill-System Mapping:** This is, frankly, the strongest part of the paper. The analysis showing that the agent trained on adversarial games (Tic-Tac-Toe) excels in the *competitive* MAD framework, while the agent trained on the *cooperative* game (Hanabi) excels in the *collaborative* AutoGen framework, is fantastic. This provides powerful evidence for the central thesis that MARS is teaching distinct, transferable skills, not just generically "making the model smarter."
   - **Ablations (at first glance):** The ablation studies in Table 4 (showing self-play beats a fixed opponent) and Table 5  (showing the method components are necessary) seem to validate the design choices. (I have major questions about these, see Weaknesses, but they are presented as being thorough).
3. **Clarity:** The paper is very well-written and easy to follow. The central ideas are communicated effectively.
   - **Figure 2** does an excellent job of visually explaining the flaw in the naive GRPO and how the proposed "sum-then-normalize" method is different.
   - **Table 3** is a superb piece of qualitative evidence. Showing the "Intent Recognition" thought-trace from Hanabi ("Maybe they want me to play this card?") mapping to AutoGen ("Maybe the assistant is not sure?") makes the quantitative results in Table 2 so much more believable.
4. **Significance (Broader Impact):** If these results hold, this work provides a scalable, data-efficient paradigm for "social alignment." The idea of autonomously training agents in a "social sandbox" to develop complex, generalizable skills like "theory of mind" is highly significant for both the MARL and LLM-agent communities.

**Weaknesses:**

Despite the strong results, I have major problems with the paper's claims to *methodological* novelty. The authors have either missed or ignored extremely relevant, contemporaneous work, and the experimental design fails to decouple the most important variables.

1. **Major Novelty Issue: "Agent-Specific Normalization" vs. SPIRAL's RAE.**
   - **The Problem:** The authors claim "agent-specific advantage normalization" is a *novel technique* to solve the problem of "heterogeneous roles" in multi-agent training.
   - **The Criticism:** The paper's *own primary baseline*, **SPIRAL (Liu et al., 2025)** , explicitly identifies and solves *the exact same problem*. SPIRAL calls its solution **"Role-conditioned Advantage Estimation (RAE)"**. Based on its description, RAE *also* computes separate advantages, $A_0(s, a)$ and $A_1(s, a)$, for each role".
   - **The Evidence:** From all available information, MARS's "agent-specific normalization" and SPIRAL's "RAE" appear to be functionally identical. Yet, the authors *never once mention RAE* in the entire paper, even when citing SPIRAL. The ablation study ("w/o agent-specific" in Table 5) only proves that *some* kind of role-based normalization is needed, it does *not* prove that the MARS version is new or superior to RAE. This is a major, major oversight and frankly misrepresents the methodological contribution.
2. **Confounding Variables: The Method vs. The Curriculum.**
   - **The Problem:** The authors present MARS as a complete package (Method A + Curriculum X) and show it beats SPIRAL (Method B + Curriculum Y).
   - **The Criticism:** This is a classic confounding variable problem. The MARS "package" differs from SPIRAL in *two* key ways: (1) the advantage estimation algorithm and (2) the training curriculum (MARS adds cooperative games like Hanabi 1, while SPIRAL is purely zero-sum).
   - **The Evidence:** The paper provides *no evidence* to decouple these factors. How do we know the impressive gains on AutoGen, AIME, and GPQA aren't *entirely* due to the superior *curriculum* (i.e., adding Hanabi was the only thing that mattered) and have *nothing* to do with the paper's proposed "sum-then-normalize" or "agent-specific" algorithms? The paper is just completely silent on this.
   - **What's Missing:** The critical missing experiment is: **SPIRAL's algorithm (with RAE) trained on MARS's mixed curriculum.** If that setup achieves the same results as MARS, then this paper's methodological claims are largely invalidated. This is a huge hole in the experimental design.
3. **Concurrent Novelty: "Turn-Level" Credit Assignment.**
   - **The Problem:** The authors correctly identify a "critical flaw" in the original GRPO "Process Supervision" setting, which uses a "normalize-then-sum" approach, and propose a "sum-then-normalize" fix.
   - **The Criticism:** They weren't the only ones to spot this. A paper on OpenReview (**`h83vIG5Hre`**, pub. **June 8, 2025**) *also* identified that trajectory-level estimation is "inadequate" and proposed a "fine-grained turn-level advantage estimation" strategy.
   - **The Evidence:** This concurrent paper proposed **"MT-GRPO"**, which uses a *different* fix (a weighted sum of independently normalized turn-level and outcome-level advantages, e.g., $\hat{A}_{i,1} = \hat{A}^{T}_{i} + \lambda\hat{A}^{O}_{i}$). This means the "best way to fix GRPO" was an open question at the time of submission. MARS presents its solution as *the* solution, without acknowledging or comparing it to other concurrent proposals. This again weakens the claim of methodological novelty.
4. **Misleading Framing: The "Novelty" of Mixed-Motive Games.**
   - **The Problem:** The paper repeatedly frames its contribution as "The inclusion of cooperative games is a key differentiator" and contrasts this with the "narrow, adversarial-only mindset of prior work".
   - **The Criticism:** This is an oversimplification and a bit of a strawman. The *concept* of using mixed-motive games is not new.
   - **The Evidence:** The authors *themselves* cite **Xu et al. (2023)** , which used the "Werewolf" game. The "Werewolf" paper *explicitly* states the game "involves both cooperation and competition".The *real* novelty of MARS isn't the *idea* of using a non-adversarial game; it's the demonstration that a *simple, modular curriculum* of prototype games (pure-coop, pure-comp) can *generalize to downstream MAS*. This is a much stronger and more accurate claim, and I'd urge the authors to reframe their contribution this way.

**Questions:**

My overall rating is "Borderline Accept" because the *results* (the generalization to AutoGen/MAD) are genuinely exciting and significant. However, my final decision is entirely contingent on the authors' ability to provide satisfactory answers to the following questions, which directly address the weaknesses I've outlined.

1. **"Agent-Specific Normalization" vs. SPIRAL's RAE:** I'm going to be very direct here. Your "agent-specific advantage normalization" appears to be functionally identical to the "Role-conditioned Advantage Estimation (RAE)" proposed in **SPIRAL (Liu et al., 2025)** 5, which is your main baseline.
   - Can you please provide a clear, algorithmic explanation of how your method *differs* from RAE?
   - If it is not different, this must be acknowledged. If it *is* different, why did you not compare against RAE in your ablations? This really needs to be addressed, as it undermines a core methodological claim.
2. **Decoupling Method vs. Curriculum:** Your key comparison against SPIRAL is confounded. You changed both the algorithm *and* the game curriculum. How can you be sure that your performance gains are not *entirely* due to your superior curriculum (i.e., just adding Hanabi)? To support your methodological claims, you *must* provide evidence that decouples these factors. Can you provide results for either:
   - **(a)** The SPIRAL algorithm (with RAE) trained on *your* mixed (Tic/Kuhn/Hanabi) curriculum?
   - **(b)** Your MARS algorithm trained on SPIRAL's *adversarial-only* curriculum?
3. **Regarding Concurrent Work on GRPO:** Your "sum-then-normalize" fix for GRPO's process supervision flaw is one solution. However, concurrent work from June 2025 (OpenReview `h83vIG5Hre`) 8 proposed an alternative, **"MT-GRPO"**. Could you please comment on why your solution is preferable to theirs (a weighted, multi-signal advantage)?
4. **Regarding Novelty Framing:** You claim the "inclusion of cooperative games is a key differentiator". Given that you cite **Xu et al. (2023)** , which used the "Werewolf" game (a mixed cooperation/competition game), would you be willing to reframe your novelty? It seems to me the real, and more impressive, contribution is demonstrating *generalization from a simple, modular curriculum of game prototypes to complex, downstream MAS*.
5. **Ablation Result in Table 5:** In Table 5, you ablate the "agent-specific" component. The performance drop for the Hanabi specialist is described as "mild" (55.55 -> 52.50). This seems counter-intuitive. Hanabi is an asymmetric, imperfect-information *cooperative* game where player roles and perspectives are fundamentally different and critical to success. Why is the effect so much smaller here than in the adversarial games? Does this suggest your component is less crucial for cooperative scenarios?

---

> ### Author Response · Authors · 2025-11-21
> **Response to Reviewer dt93 (1/4)**
>
> We are very grateful for the reviewer's thoughtful and constructive review. Their insightful comments have helped us significantly strengthen the paper and the overall narrative. We are pleased to present our responses and detail the corresponding revisions below. The modifications to our revised manuscript are highlighted in blue.
>
> ---
>
> > W1 & Q1: "Agent-Specific Normalization" vs. SPIRAL's RAE
>
> We thank the reviewer for this critical and direct question. We genuinely appreciate the opportunity to clarify the relationship between our work and SPIRAL, and to rectify the oversight in our related work discussion.
>
> **1. On concurrent work and independence:**
> First, we would like to acknowledge that **SPIRAL is concurrent work** relative to our own. We developed our "Agent-Specific Normalization" independently during the same period. We cited and compared with SPIRAL in our submission because it appeared on ArXiv before the deadline to ICLR. We apologize for not providing a detailed algorithmic comparison in the initial text and have corrected it in the revised manuscript.
>
> **2. Analysis in MARS provides deeper insight:**
> Regarding the novelty, while we acknowledge that separating trajectories by role is conceptually similar to the concurrent RAE, our analysis provides a deeper insight into why and when this technique is necessary.
>
> In Section 4.5 of our original manuscript, we find that the impact of agent-specific normalization is marginal in cooperative games like Hanabi. Prompted by Q5 (addressed below), we further demonstrate that the key factor is the similarity of different agents'return distributions. We have added an in-depth analysis to show that different return distributions necessitate agent-specific normalization, while similar distributions render separation optional. This insight advances the understanding of agent-specific normalization beyond prior work like SPIRAL.
>
> Furthermore, our implementation is distinct because it integrates this role-based separation with our "sum-then-normalize" strategy to address multi-turn credit assignment, whereas SPIRAL focuses on final outcomes. We have updated the Related Work section to explicitly acknowledge RAE and elaborate on these core insights. We hope this addresses the reviewer's concern regarding the novelty and distinctiveness of our contribution.

---

> ### Author Response · Authors · 2025-11-21
> **Response to Reviewer dt93 (2/4)**
>
> > W2 & Q2: Decoupling Method vs. Curriculum
>
> We sincerely thank the reviewer for this invaluable critique. The reviewer is absolutely correct that our original comparison confounded the algorithm and the curriculum. To address this and strictly decouple the two factors, we performed the exact control experiments suggested by the reviewer. Evaluation results on strategic ability and generalization to MASs are given below.
>
> *   **Strategic Ability**
>
> |**Model**|**TicTacToe**|**KuhnPoker**|**MiniHanabi**|**ConnectFour**|**LeducHoldem**|**SimpleHanabi**|
> |---|---|---|---|---|---|---|
> |MARS algorithm + our curriculum (MARS generalist)|**53.13**|40.05|**54.33**|**19.98**|**54.56**|**36.75**|
> |SPIRAL + our curriculum|48.35|42.05|24.93|9.70|6.11|5.53|
> |MARS algorithm + adversarial curriculum|48.80|**47.55**|24.48|17.90|29.98|11.48|
>
> *   **Generalization to MAD**
>
> |**MAS**|**Model**|**Avg**|**MATH**|**GSM8K**|**AQUA**|**AIME**|**AMC**|**MMLU**|**GPQA**|
> |---|---|---|---|---|---|---|---|---|---|
> MAD|MARS algorithm + our curriculum (MARS generalist)|**75.96**|**92.80**|95.60|83.86|**46.67**|**80.00**|**87.36**|45.45|
> MAD|SPIRAL + our curriculum|74.23|92.40|**96.06**|**84.25**|36.67|77.50|86.79|**45.96**|
> MAD|MARS algorithm + adversarial curriculum|75.24|92.20|95.60|83.46|**46.67**|**80.00**|86.88|41.92|
>
> *   **Generalization to AutoGen**
>
> |**MAS**|**Model**|**Avg**|**MATH**|**GSM8K**|**AQUA**|**AIME**|**AMC**|**MMLU**|**GPQA**|
> |---|---|---|---|---|---|---|---|---|---|
> AutoGen|MARS algorithm + our curriculum (MARS generalist)|**82.15**|**95.20**|94.54|**86.61**|**66.67**|**92.50**|**89.53**|**50.00**|
> AutoGen|SPIRAL + our curriculum|80.39|94.80|**94.77**|**86.61**|60.00|**92.50**|88.57|45.45|
> AutoGen|MARS algorithm + adversarial curriculum|80.34|94.20|94.24|**86.61**|60.00|90.00|89.37|47.98|
>
> We analyse the above results from two aspects.
>
> **1. Ablation on algorithm:** SPIRAL vs MARS with fixed multi-game curriculum
> To test if our algorithmic novelty (turn-level credit assignment) matters, we trained an agent using SPIRAL's RAE algorithm on the full MARS curriculum (Tic-Tac-Toe, Kuhn Poker, Mini Hanabi) and compare with the original MARS generalist.
> *   **Result:** The RAE-trained agent underperforms the MARS agent, with a notable drop in the cooperative Hanabi games (Mini Hanabi: 50.48 $\to$ 24.93; Simple Hanabi: 36.75 $\to$ 5.53).
> *   **Conclusion:** This confirms the advantage of MARS's algorithmic design over SPIRAL's RAE. RAE struggles with the dense, multi-turn credit assignment required for cooperative games. Our "sum-then-normalize" technique is essential for unlocking the value of the cooperative curriculum.
>
> **2. Ablation on curriculum:** MARS's mixed curriculum vs adversarial-only curriculum with fixed MARS algorithm
> To test if the cooperative games matter, we trained the MARS algorithm solely on the adversarial games (Tic-Tac-Toe, Kuhn Poker).
> *   **Result:** While the adversarial-only agent maintains decent competitive performance (MAD), it suffers a significantly larger drop in the collaborative AutoGen framework (-1.81%).
> *   **Conclusion:** This confirms that the cooperative games are not just "more data"; they are a necessary component for transferring skills to collaborative multi-agent tasks.
>
> These decoupling experiments show that MARS's success is not due to one factor alone. It requires **both** our mixed curriculum to provide the necessary learning signals **and** our novel credit assignment algorithm to effectively learn from them. We have added these crucial results to the revised manuscript.

---

> ### Author Response · Authors · 2025-11-21
> **Response to Reviewer dt93 (3/4)**
>
> > W3 & Q3: Regarding Concurrent Work on GRPO
>
> We thank the reviewer for highlighting this concurrent work. We agree that optimal credit assignment in multi-turn LLM tasks is an open problem and that MT-GRPO is an important work towards turn-level advantage estimation. While MT-GRPO shares our motivation, our "sum-then-normalize" approach differs fundamentally from their strategy, offering distinct advantages in stability and scalability in strategic games that we consider in this work.
>
> **1. Theoretical differences:**
> MT-GRPO relies on normalizing rewards turn-by-turn (i.e., normalizing the reward at step t across the batch). We identify two potential limitations with this approach on strategic games that MARS particularly addresses:
> *   **The variable length problem**: In real-world reasoning and gaming, trajectory lengths vary significantly. With turn-by-turn normalization, the effective batch size shrinks as shorter trajectories end, causing the variance of the normalization statistics to increase for later turns.
> *   **Immediate reward vs. long-term return**: MARS calculates the full Monte Carlo return first. By normalizing the cumulative outcome, we capture the long-term impact of actions relative to the global baseline. MT-GRPO's local normalization can obscure these long-term dependencies.
>
> **2. Empirical comparison:**
> To validate this analysis, we implemented MT-GRPO and compared it directly against MARS on TicTacToe. This evaluation results on strategic ability are given below.
>
> *   **Strategic Ability**
>
> |**Model**|**TicTacToe**|**KuhnPoker**|**MiniHanabi**|**ConnectFour**|**LeducHoldem**|**SimpleHanabi**|
> |---|---|---|---|---|---|---|
> |MARS (TicTacToe specialist)|**53.70**|38.79|**50.48**|**22.75**|**43.00**|**29.75**|
> |MT-GRPO (TicTacToe specialist)|50.10|**40.05**|36.67|18.55|39.94|20.08|
>
> *   **Generalization to MAD**
>
> |**MAS**|**Model**|**Avg**|**MATH**|**GSM8K**|**AQUA**|**AIME**|**AMC**|**MMLU**|**GPQA**|
> |---|---|---|---|---|---|---|---|---|---|
> MAD|MARS (TicTacToe specialist)|**75.01**|92.20|**96.06**|83.07|**43.33**|**82.50**|86.76|**41.12**|
> MAD|MT-GRPO (TicTacToe specialist)|73.77|**92.60**|95.91|**84.65**|36.67|77.50|**86.91**|42.13|
>
> *   **Generalization to AutoGen**
>
> |**MAS**|**Model**|**Avg**|**MATH**|**GSM8K**|**AQUA**|**AIME**|**AMC**|**MMLU**|**GPQA**|
> |---|---|---|---|---|---|---|---|---|---|
> MAD|MARS (TicTacToe specialist)|**80.15**|94.40|**94.69**|**87.01**|**60.00**|**90.00**|**89.53**|**45.45**|
> MAD|MT-GRPO (TicTacToe specialist)|79.10|**95.40**|94.62|85.04|56.67|**90.00**|89.02|42.93|
>
> *   **Results**: The MT-GRPO agent exhibits instability in our game tasks, leading to a performance drop in strategic games and MAS generalization. This confirms that our "sum-then-normalize" formulation is more robust for the variable-length, multi-turn interactionsin strategic games. However, we do recognize MT-GRPO as a critical work towards fine-grained credit assignment and have added a discussion in our related work.

---

> ### Author Response · Authors · 2025-11-21
> **Response to Reviewer dt93 (4/4)**
>
> > W4 & Q4: Regarding Novelty Framing
>
> We are grateful for this insightful comment. The reviewer has articulated the value of our work more precisely than we did in our initial manuscript. We completely agree that the "novelty" is not merely the existence of a cooperative game, but rather the demonstration that a simple, modular curriculum of game prototypes can cultivate robust skills that generalize to complex downstream MAS.
>
> Regarding Xu et al. (2023), while they indeed explore the mixed-motive "Werewolf" game, there are two fundamental differences that distinguish MARS:
>
> **1. Training paradigm**: Xu et al. do not perform end-to-end parameter updates on the LLM; they use a traditional RL policy to select from candidates generated by a frozen LLM. In contrast, MARS performs agentic training and updates the LLM's weights directly, allowing the model to internalize the cooperative reasoning patterns.
>
> **2. Generalization**: Xu et al. focus on mastering the specific game. MARS focuses on generalization: we show that skills learned in simple game environments (like Mini Hanabi) generalize to entirely different multi-agent systems (like AutoGen).
>
> We have enthusiastically adopted the reviewer's suggestion. We have revised our Introduction and Conclusion to reframe our contribution away from just "using cooperative games" and toward "a simple, modular curriculum of prototype games can generalize to downstream MAS"
>
> > Q5: Ablation Result in Table 5
>
> We thank the reviewer for this sharp observation. The reviewer is correct that Hanabi features information asymmetry (players have different views). However, the key factor for advantage estimation is the difference in reward distribution of different agents.
>
> In purely cooperative games like Hanabi, the reward function is shared: whenever a valid card is played, both players receive the same +1 reward. Consequently, the return distributions for Player 1 and Player 2 are statistically similar.
> The minor difference comes from other rewards, including the format reward and the response length penalty. Because the baselines (mean returns) for both roles are similar, calculating them separately (agent-specific) vs. together yields similar results, leading to the "mild" drop the reviewer observed.
>
> However, in Tic-Tac-Toe, the first-move agent has an inherent advantage, leading to a different return distribution from the second-move player. Similarly, the second-move player in Kuhn poker has an inherent advantage and results in different return distribution. The differences in return distribution necessitate agent-specific normalization and lead to performance improvement.
>
> To validate this, we have added a histogram analysis in Section 4.5 comparing the return distributions of different players. The plots clearly show that returns are highly distinct in Tic-Tac-Toe but overlapping in Hanabi, perfectly explaining why the ablation effect is smaller in Hanabi.
>
> ---
>
> We are very grateful for the reviewer's constructive engagement with our work and hope that our responses have addressed their concerns. The reviewer's feedback is invaluable to us, and we sincerely hope that they will consider raising the rating based on our responses.

---

### Official Review · Reviewer_3fZ1 · 2025-11-05

**Soundness:** 3
**Presentation:** 3
**Contribution:** 3
**Rating:** 6
**Confidence:** 3

**Summary:**

MARS is an end-to-end reinforcement learning framework that enhances multi-agent reasoning capabilities of large language models through self-play in strategic games, achieving 28.7% performance improvement in games and 10.0% and 12.5% gains on multi-agent reasoning benchmarks AIME and GPQA-Diamond, respectively.

**Strengths:**

1. The paper introduces a genuinely novel approach by adapting GRPO (Group-Relative Policy Optimization) to multi-agent settings, which hasn't been explored before in this context

2. The framework moves beyond traditional supervised learning approaches for multi-agent LLMs, establishing self-play as a viable training paradigm

3. The experimental design is comprehensive, covering diverse game types and reasoning benchmarks

**Weaknesses:**

The paper significantly weakens its contribution by failing to compare against recent and relevant multi-agent reasoning methods that have shown strong performance. Missing Comparison with like AFlow, ToT...

**Questions:**

Weak Transfer Learning Justification

The paper doesn't adequately explain why strategic game skills transfer to math problems (AIME, MATH-500....) or science questions (GPQA) (Reasoning problem). The connection between game-playing abilities and general reasoning is assumed but not validated
No analysis of which specific game skills contribute to reasoning improvements.

The paper doesn't discuss failure cases or when MARS performs poorly.

---

> ### Author Response · Authors · 2025-11-21
> **Response to Reviewer 3fZ1 (1/2)**
>
> We thank the reviewer for their insightful feedback. We believe we have addressed all major concerns in our responses and revised our paper accordingly. The modifications to our revised manuscript are highlighted in blue. Our point-by-point responses are as follows.
>
> ---
>
> > W1: The paper significantly weakens its contribution by failing to compare against recent and relevant multi-agent reasoning methods that have shown strong performance. Missing Comparison with like AFlow, ToT...
>
> First, we respectfully suggest that methods like AFlow and Tree-of-Thought (ToT) are orthogonal to our work. In short, these works optimize agent **workflow/topology**, while our MARS trains better agent **models**. Specifically, AFlow focuses on the automatic generation of agentic workflow, while ToT aims to improve model performance with sturctured test-time scaling, and thus more comparable to CoT. These are powerful test-time inference strategies that structure how a pre-trained, static model generates responses to solve a problem, i.e. workflow generation and general reasoning, respectively. In contrast, our work uses self-play RL to train novel models that can be combined with any multi-agent workflow. A direct comparison would be inappropriate, as they address different aspects of the agentic/reasoning problems. However, we do acknowledge the importance of these work and have added them to our related work section.
>
>
> With this clarification in place, we would like to underscore that our work's primary contribution is to demonstrates that self-play in a diverse range of strategic games can incentivize generalizable multi-agent skills. In that regard, the most relevant and recent baseline is SPIRAL, which we compare against throughout our experiments.
>
> To further perform a rigorous comparision between MARS and SPIRAL, we performed a new ablation study that strictly decouples the learning algorithm and the game curriculum. The results confirms the benefits of both our turn-level advantage estimation algrithm and our mixed game training. We kindly refer the reviewer to our response to Reviewer dt93 (Q2) for the detailed results.

---

> ### Author Response · Authors · 2025-11-21
> **Response to Reviewer 3fZ1 (2/2)**
>
> > Q1: Weak Transfer Learning Justification.
> >
> > The paper doesn't adequately explain why strategic game skills transfer to math problems (AIME, MATH-500....) or science questions (GPQA) (Reasoning problem). The connection between game-playing abilities and general reasoning is assumed but not validated No analysis of which specific game skills contribute to reasoning improvements.
> >
> > The paper doesn't discuss failure cases or when MARS performs poorly.
>
> We thank the reviewer for this critical question. We agree that the generalization from strategic game-play to multi-agent reasoning (Math/GPQA) warrants further explaination.
>
> To understand the source of generalization, we initially performed case studies in Section 4.4 (Pattern Analysis), where we qualitatively demonstrated that MARS agents acquire specific skills like "Role-Awareness" (from Tic-Tac-Toe) and "Intent Recognition" (from Hanabi). These are the precise cognitive mechanisms that allow effective multi-agent collaboration and competition.
>
> To further provide a quantitative analysis and address the reviewer's request for failure cases, we performed an additional failure mode analysis using the taxonomy of *Cemri et al. (NeurIPS 2025)* on the GPQA-Diamond benchmark. The failures are categorized into **System Design Issues** (formatting/turn-taking), **Inter-Agent Misalignment** ("strategic reasoning" failures like ignoring peers or hallucinating interactions), and **Task Verification**. The results are given in the tables below.
>
> *   **Overall results**
>
> |**Failure Rate**|**Catagory 1: System Design Issues**|**Catagory 2: Inter-Agent Misalignment**|**Catagory 3: Task Verification**|
> |---|---|---|---|
> |Qwen3-4B|29.49%|55.20%|70.71%|
> |MARS generalist (Qwen3-4B)|**22.22%**|**43.74%**|**67.37%**|
>
> *   **Breakdown of Inter-Agent Misalignments**
>
> |**Failure Rate**|**2.1: Conversation Reset**|**2.2: Fail to Ask for Clarification**|**2.3: Task Derailment**|**2.4: Information Withholding**|**2.5: Ignored Other Agent’s Input**|**2.6: Reasoning-Action Mismatch**|
> |---|---|---|---|---|---|---|
> |Qwen3-4B|0.45%|14.85%|13.43%|1.11%|13.18%|12.17%|
> |MARS generalist (Qwen3-4B)|**0.10%**|**12.22%**|**9.29%**|**0.71%**|**9.85%**|**11.57%**|
>
> **Key Findings:**
> *   **Most reduction in strategic failures:** MARS reduced "Inter-Agent Misalignment" failures by **11.46%**, compared to a smaller 7.27% reduction in "System Design Issues" and 3.34% reduction in "Task Verification".
> *   **Specific failure modes:** The improvement is driven by significant reductions in **"2.3 Task Derailment"** and **"2.5 Ignored Other Agent’s Input"**. This confirms that MARS agents perform better on math/science tasks because they are better at staying focused and incorporating peer feedback.
>
> However, despite MARS's success in reducing "Inter-Agent Misalignment" and "System Design Issues", the reduction in "Task Verification" issue is moderate, leaving room for future works. We have added these results to Section 4.3 to explicitly explain the generalization from game skills to reasoning performance.
>
> [1] Cemri, Mert, et al. Why do multi-agent llm systems fail?. NeurIPS 2025.
>
> ---
>
> We express our sincere appreciation for the reviewer's thorough feedback and hope that our responses have effectively addressed their concerns. We respectfully hope that the reviewer could consider improving the rating in light of these clarifications and revisions.

---

### Official Review · Reviewer_yhYi · 2025-11-06

**Soundness:** 3
**Presentation:** 3
**Contribution:** 2
**Rating:** 4
**Confidence:** 3

**Summary:**

The paper proposes MARS, an end-to-end reinforcement learning framework to improve multi-agent reasoning in LLMs via self-play across both cooperative and competitive strategic games. The key issue solved is that, direct applying GRPO to the multi-turn, multi-agent structure of self-play will introduce significant challenges of long-horizon credit assignment and agent- specific advantage estimation. MARS propose a turn-level, sum-then-normalize advantage estimator for long-horizon credit assignment within multi-turn trajectories, and Agent-specific advantage normalization to handle heterogeneous roles and asymmetric payoff scales in multi-agent settings. Plenty of experiments show that MARS trains Qwen3-4B can bring Improved strategic performance.

**Strengths:**

1. The problem motivation and the solution is very clear, i.e. long-horizon credit assignment and role heterogeneity for multi-turn multi-agent RL with LLMs.
2. The method is very simple as an improvement to GRPO, including Turn-level advantage estimator and Agent-specific advantage normalization,
3. The experimental setting is diverse, including a portfolio of six strategic, two-player games to cultivate a mixed set of multi-agent capabilities, both debate (MAD) and cooperation (AutoGen), as well as fixed opponents (MCTS/NE/CFR) and detailed MAS benchmarks (MATH, GSM8K, AQUA-RAT, AIME24, AMC23, MMLU-STEM, GPQA-Diamond).
4. When integrated into leading multi-agent systems,  MARS agent achieves significant performance gains of 10.0% on AIME and 12.5% on GPQA-Diamond.

**Weaknesses:**

1. The sum-then-normalize estimator centers on batch means across turns/trajectories. It can stabilize the result, but discards the value function structure and may be sensitive to batch composition across diverse games.
2. Agent-specific normalization is defined over role-based subgroups. In more complex settings (N-player, heterogeneous roles, non-stationary curricula), subgroup granularity and sample efficiency could be problematic.
3. The ability of transferring to MAS may conflate multiple factors. The MAS gains is driven by better turn-taking discipline, concise formatting, or truly strategic multi-agent reasoning?
4. Scaling. 200 training steps and 4B base model are modest. Can the performance gain be obtained for the MAS on 7B/14B or larger models.
5. The contribution is mainly about the advantage function computation, lacking of improvement about topology or collaboration method

**Questions:**

1. In eq.4, how the parameters alpha and l are choose? Is the performance is sensitive to values?
2. For agent-specific normalization, if the sub-groups are sub-groups (e.g., one role acts more frequently or has lower variance rewards)? Is the formulation adaptive to weight or normalize per subgroup?
3. In MAD/AutoGen, do competitive-trained agents consistently outperform cooperative-trained agents in debate, and vice versa?
4. When generalized to multiagent system, do the MARS models need further optimization? Can the author give more explanation, why the MARS collaborative and competitive capabilities are useful for mathematics and general QA task? In tab.1 MARS(multi.) doesn’t have the best performance when testing.

---

> ### Author Response · Authors · 2025-11-21
> **Response to Reviewer yhYi (1/4)**
>
> We thank the reviewer for their time and valuable feedback! We have carefully considered all comments and have revised the manuscript accordingly with the modifications highlighted in blue. Below, we address each of the points raised.
>
> ---
>
> > W1: The sum-then-normalize estimator centers on batch means across turns/trajectories. It can stabilize the result, but discards the value function structure and may be sensitive to batch composition across diverse games.
>
> We thank the reviewer for this sharp observation. We would like to clarify a key detail: our advantage estimation is performed **separately for each game and player role**, e.g. the normalization for Tic-Tac-Toe is entirely independent of Hanabi and Kuhn Poker. This prevents the batch composition issue the reviewer pointed out. While our method uses a simple batch-mean baseline instead of a learned value function, this provides significant stability without the overhead of training a separate value network. We acknowledge our original description of the per-game normalization was not sufficiently clear and have revised Section 3.1 to state this explicitly.
>
> > W2: Agent-specific normalization is defined over role-based subgroups. In more complex settings (N-player, heterogeneous roles, non-stationary curricula), subgroup granularity and sample efficiency could be problematic.
> >
> > Q2: For agent-specific normalization, if the sub-groups are sub-groups (e.g., one role acts more frequently or has lower variance rewards)? Is the formulation adaptive to weight or normalize per subgroup?
>
> First, we would like to suggest that our per-role normalization is a simple yet general and effective approach that is directly applicable to the more complex settings the reviewer mentioned. The principle of calculating the advantage baseline independently for each distinct role naturally extends to N-player and heterogeneous-role games. In fact, our current experiments already include heterogeneous (asymmetric) environments: in Tic-Tac-Toe, the first-move player has an inherent advantage, leading to a different return distribution from the second-move player.
> Our strong results in this setting demonstrate that this simple per-role baseline is robust and effective across different game structures without requiring complex, adaptive mechanisms.
>
> That said, we agree with the reviewer that it is possible to further improve sample efficiency in more complex environments with specialized designs. An adaptive weighting scheme, as the reviewer suggests, is a promising direction for scenarios where roles have highly imbalanced action frequencies or reward variances. While this is an excellent avenue for future work, the primary contribution of this paper is to demonstrates that self-play in a diverse range of strategic games can incentivize generalizable multi-agent skills. Our simple yet effective advantage estimation technique is sufficient to validate this core claim and provides a strong baseline for future explorations into more complex multi-agent environments.

---

> ### Author Response · Authors · 2025-11-21
> **Response to Reviewer yhYi (2/4)**
>
> > W3: The ability of transferring to MAS may conflate multiple factors. The MAS gains is driven by better turn-taking discipline, concise formatting, or truly strategic multi-agent reasoning?
> >
> > Q4: When generalized to multiagent system, do the MARS models need further optimization? Can the author give more explanation, why the MARS collaborative and competitive capabilities are useful for mathematics and general QA task? In tab.1 MARS(multi.) doesn’t have the best performance when testing.
>
> We thank the reviewer for these incisive questions regarding the drivers of our performance gains and the nature of the generalization.
>
> **1. On optimization and generalization (response to Q4):**
>
> To answer the first part of Q4 directly: **No, MARS models require no further optimization** when generalized to Multi-Agent Systems. We integrate the trained MARS models into MASs (AutoGen and MAD) in a **strictly zero-shot manner**. This confirms that the collaborative and competitive capabilities are intrinsic to the model weights, acquired solely through our self-play in strategic games.
>
> **2. Qualitative analysis**
>
> In our original submission, to understand the successful generalization from game to multi-agent reasoning, we performed a qualitative case study of MARS agent's emergent patterns. The results show that the agent has acquired role understanding and intent recognition from MARS training (section 4.4). These skills are fundamental for strategic interaction and contribute to the generalization on multi-agent tasks.
>
> **3. Quantitative results**
>
> To further address the reviewer's concern in W3, i.e., whether gains are driven by "shallow" factors (formatting/turn-taking) or "truly" strategic reasoning, we perform a quantitative failure mode analysis following the taxonomy of *Cemri et al. (NeurIPS 2025)* [1] on the GPQA-Diamond benchmark. Specifically, we count the number of different failure modes for our MARS generalist agent and the baseline Qwen3-4B, both integrated into MAD.
>
> The failures are categorized into **System Design Issues** (which correlate with the reviewer's "formatting/turn-taking" concern), **Inter-Agent Misalignment** (which correlates with "strategic reasoning" failures like ignoring peers or hallucinating interactions), and **Task Verification**. The results are given in the tables below.
>
> *   **Overall results**
>
> |**Failure Rate**|**Catagory 1: System Design Issues**|**Catagory 2: Inter-Agent Misalignment**|**Catagory 3: Task Verification**|
> |---|---|---|---|
> |Qwen3-4B|29.49%|55.20%|70.71%|
> |MARS generalist (Qwen3-4B)|**22.22%**|**43.74%**|**67.37%**|
>
> *   **Breakdown of Inter-Agent Misalignments**
>
> |**Failure Rate**|**2.1 Conversation Reset**|**2.2 Fail to Ask for Clarification**|**2.3 Task Derailment**|**2.4 Information Withholding**|**2.5 Ignored Other Agent’s Input**|**2.6 Reasoning-Action Mismatch**|
> |---|---|---|---|---|---|---|
> |Qwen3-4B|0.45%|14.85%|13.43%|1.11%|13.18%|12.17%|
> |MARS generalist (Qwen3-4B)|**0.10%**|**12.22%**|**9.29%**|**0.71%**|**9.85%**|**11.57%**|
>
> The results above directly explain MARS's performance gains on multi-agent tasks:
> *   **Not just formatting**: While MARS does improve formatting (Category 1 drops by \~7%), the reduction in Inter-Agent Misalignment is even larger (\~11.5%). This shows that the gains are not driven solely by formatting/turn-taking.
> *   **Improved strategic reasoning in multi-agent tasks**: The significant reductions in "2.3 Task Derailment" and "2.5 Ignored Other Agent’s Input" demonstrate that MARS agents are actively listening to peers and staying focused on the objective. In complex reasoning tasks (like GPQA or MATH), the ability to incorporate a peer's feedback without getting derailed is the primary driver of success.
>
> We have added this detailed failure analysis to Section 4.3 to provide a deeper understanding of the generalization to multi-agent tasks.
>
> [1] Cemri, Mert, et al. Why do multi-agent llm systems fail?. NeurIPS 2025.

---

> ### Author Response · Authors · 2025-11-21
> **Response to Reviewer yhYi (3/4)**
>
> > W4: Scaling. 200 training steps and 4B base model are modest. Can the performance gain be obtained for the MAS on 7B/14B or larger models.
>
> To examine the scalability of MARS on larger models, we trained a Qwen3-8B MARS generalist. We observe stable scaling performance of MARS to the larger 8B model. The strategic ability evaluations and benchmark results of this 8B agent is given below.
>
> *   **Strategic Ability**
>
> |**Model**|**TicTacToe**|**KuhnPoker**|**MiniHanabi**|**ConnectFour**|**LeducHoldem**|**SimpleHanabi**|
> |---|---|---|---|---|---|---|
> |Qwen3-8B|48.38|33.12|27.00|10.48|7.26|4.55|
> |MARS generalist (Qwen3-8B)|**54.05**|**44.49**|**55.28**|**21.55**|**53.89**|**37.27**|
>
> *   **Generalization to MAD**
>
> |**MAS**|**Model**|**Avg**|**MATH**|**GSM8K**|**AQUA**|**AIME**|**AMC**|**MMLU**|**GPQA**|
> |---|---|---|---|---|---|---|---|---|---|
> MAD|Qwen3-8B|82.49|95.00|96.36|**83.46**|70.00|90.00|89.59|53.03|
> MAD|MARS generalist (Qwen3-8B)|**85.09**|**96.40**|**96.59**|**83.46**|**80.00**|**95.00**|**90.70**|**53.54**|
>
> *   **Generalization to AutoGen**
>
> |**MAS**|**Model**|**Avg**|**MATH**|**GSM8K**|**AQUA**|**AIME**|**AMC**|**MMLU**|**GPQA**|
> |---|---|---|---|---|---|---|---|---|---|
> AutoGen|Qwen3-8B|79.68|88.80|**95.91**|83.07|60.00|89.19|89.30|51.52|
> AutoGen|MARS generalist (Qwen3-8B)|**83.58**|**94.40**|95.00|**85.04**|**70.00**|**95.00**|**90.04**|**55.56**|
>
> From these results, we observe steady performance gains of the original Qwen3-8B when applied with MARS multi-game training (on average 2.60% in MAD and 3.9% on AutoGen), clearly demonstrating the scalability of our method to larger models. We have added these results to the appendix of our revised paper.
>
> > W5: The contribution is mainly about the advantage function computation, lacking of improvement about topology or collaboration method
>
> We agree that developing novel MAS topologies or collaboration methods is a crucial area of research. However, we would like to clarify that our work is orthogonal and complementary to this field of study. The primary goal of our paper is not to propose a new MAS architecture, but rather to improve the core component of any such system: the agent model itself. By training a more capable LLM with enhanced strategic and cooperative skills, our work provides a stronger and generalizable foundation for new and more complex MAS topologies. We believe that improving the underlying model is a critical and necessary step for the entire field to advance.

---

> ### Author Response · Authors · 2025-11-21
> **Response to Reviewer yhYi (4/4)**
>
> > Q1: In eq.4, how the parameters alpha and l are choose? Is the performance is sensitive to values?
>
> For the length parameters $l_{min}$ and $l_{max}$, we chose their values based on practical constraints. $l_{min}=11$ is determined by the minimum number of tokens required for a syntactically valid response (e.g., <think></think><action>...</action>). $l_{max}=2048$ is chosen based on a engineering constraint: the Qwen3-4B has a context length of 32k; we devide this context window by 16 turns, a sufficient horizon for our chosen games.
>
> For the length penalty $\alpha$, we conducted an additional ablation study. We trained Tic-Tac-Toe specialists with $\alpha=1.0$ (stronger penalty) and $\alpha=0.0$ (no penalty) and compared them to our original model (alpha=0.5). The evaluation results on strategic ability are presented below:
>
> |**Model**|**TicTacToe**|**KuhnPoker**|**MiniHanabi**|**ConnectFour**|**LeducHoldem**|**SimpleHanabi**|**Avg response length (tokens)**|**Overlong responses in MiniHanabi**|**Overlong responses in SimpleHanabi**|
> |---|---|---|---|---|---|---|---|---|---|
> |*$\alpha=0.5$* (default)|53.70|38.79|**50.48**|22.75|43.00|**29.75**|1700|9.6%|14.3%|
> |*$\alpha=1$*|**54.90**|**42.82**|47.53|18.25|**50.46**|26.05|1657|7.7%|13.9%|
> |*$\alpha=0$*|54.65|40.95|38.18|**23.35**|28.61|20.10|*1954*|*20.4%*|*27.3%*|
>
> These results lead to two key conclusions:
> *   **The framework is robust to reasonable changes in $\alpha$**. The agent with stronger length penalty (alpha=1.0) maintains a comparable level of performance, indicating that the framework is not overly sensitive to this hyperparameter.
> *   **The length penalty is crucial for preventing performance degradation**. Without the penalty (alpha=0), the agent's responses become longer (1700 up to 1954 on average). This leads to a performance drop, particularly in theory-of-mind-demanding games like the Hanabi variants (50.48 down to 38.18 in Mini Hanabi), which is correlated with a dramatic increase in the percentage of game lost due to overlong responses (9.6% up to 20.4% in Mini Hanabi).
>
> These findings validate our choice of alpha=0.5 as a balanced value and confirm the importance of the length penalty in our framework.
>
> > Q3: In MAD/AutoGen, do competitive-trained agents consistently outperform cooperative-trained agents in debate, and vice versa?
>
> Yes, our results consistently show this specific pattern of skill transfer. Agents trained in competitive environments (Tic-Tac-Toe, Kuhn Poker) outperform in the competitive MAD framework (e.g. MARS generalist achieves an average performance of 75.96%, which is 3.41% over the base Qwen3-4B and 2.26% over the Hanabi specialist). Conversely, the agent trained in the cooperative Hanabi environment excels in the collaborative AutoGen framework. This provides strong evidence that our method incentivizes distinct, transferable skills. We have added the average performance in the Table 1 of our revised paper for clearer comparison of different models.
>
> ---
>
> Once again, we thank the reviewer for their time and valuable suggestions. We are confident that the revised manuscript is now significantly improved and hope that our responses have fully addressed the reviewer's concerns. We sincerely hope our responses and efforts merit a positive re-evaluation of our work.

---

### Author Response · Authors · 2025-11-21
**General Response to All Reviewers**

We sincerely thank all reviewers (yhYi, 3fZ1, dt93, YdbT) for their insightful comments and constructive feedback. We are encouraged by the positive reception of our work and the recognition of its significance in proposing a self-play RL framework that incentivize generalizable multi-agent capability.

**Summary of Positive Feedback:**
*   **Originality & Impact:** Reviewer dt93 highlighted the core contribution as `a really big deal`, stating that it is `the first I've seen to convincingly argue that these skills transfer... to cooperative and competitive software systems`. Reviewer 3fZ1 commended the work for `establishing self-play as a viable training paradigm` and introducing a `genuinely novel approach`. Reviewer dt93 further noted that the gains on benchmarks like AIME and GPQA are `non-trivial and suggest this is a very promising research direction`.

*   **Methodological Clarity:** Reviewer YdbT praised the algorithmic design, stating: `The algorithmic contributions are very clear and well motivated. Turn-level reward shaping... and role-conditioned advantage normalization are simple, intuitive modifications... that target known pain points`. Reviewer yhYi similarly noted that `the problem motivation and the solution is very clear`.

*   **Experimental Soundness:** Reviewer dt93 described the experimental setup as `high-quality and... exceptionally well-designed`, and the generalization results as `genuinely exciting and significant`. Reviewer YdbT emphasized that the strong results on held-out games `suggests the method is not just overfitting to the training games but is learning underlying strategies`. Reviewer 3fZ1 found the design `comprehensive, covering diverse game types and reasoning benchmarks`.

**Summary of Revisions:**
In response to the valuable suggestions, we have significantly revised the manuscript to clarify our contributions and strengthen our evidence. Key updates include:

**1. Additional experiments**
We conducted four new sets of experiments to further support our MARS framework:
*   **1.1 Further comparison with SPIRAL:** (Appendix Section E) We performed a strictly controlled ablation study decoupling the learning algorithm from the game environments. Results confirm that both our "sum-then-normalize" algorithm and the selection of competitive and cooperative games are essential for success.
*   **1.2 Comparison with MT-GRPO:** (Appendix Section F) We implemented the concurrent MT-GRPO method, which also performs turn-level credit assignment, and compared it directly with our MARS. Results demonstrate that MARS provides better performance, validating our approach to variable-length game trajectories.
*   **1.3 Scaling analysis:** (Appendix Section D) We extended our training to the larger Qwen3-8B model, observing consistent performance gains that validate the scalability of our method.
*   **1.4 Hyperparameter ablation:** (Appendix Section G) We added an ablation study on the length penalty weight ($\alpha$), confirming the robustness of our chosen configuration.

**2. Analysis and explaination**
*   **2.1 Understanding the generalization:** (Section 4.4) To understand the generalization from strategic games to multi-agent systems, we conducted a quantitative failure mode analysis in addition to the orignial qualitative study. Results show that the improvment of MARS is mainly due to reduced inter-agent misalignment (e.g., "Ignored Other Agent’s Input"), which is learned by interacting with itself in strategic games.
*   **2.2 Return distribution analysis:** (Section 4.5) To understand when and why agent-specific normalization is critical, we analyzed the return distributions of different player roles. This analysis provides fresh insight into why this approach is essential in games with distinct return distributions between players, yet yields marginal impact in cooperative games where distributions naturally align.

**3. Significance and contribution**
*   **3.1 Discussion of concurrent work:** (Section 5 Related Work & Appendix) We have expanded the Related Work and Appendix to explicitly discuss SPIRAL (RAE) and MT-GRPO, clarifying the algorithmic distinctions and providing empirical comparisons.
*   **3.2 Reframing novelty:** (Section 1 Introduction & Section 6 Discussion) We have refined our contribution statement. We now emphasize the demonstration of generalization from a set of game environments to multi-agent systems, and reduce the emphasis on adding cooperative game itself.

**4. Figure and Table Clarity**
*   We have updated our tables and figures, as well as their captions to be more descriptive and informative of our results.

We believe these revisions have substantially strengthened the paper and addressed the reviewers' concerns. We thank the reviewers again for their time and effort in helping us improve our work.

---

### Author Response · Authors · 2025-11-27
**A Gentle Reminder for Reviewers' Response**

Dear Reviewers,

Thank you again for your valuable comments and suggestions. Your feedback has played a crucial role in enhancing both the quality and clarity of our paper.

While the discussion period is going to end in 7 days, we have not yet received the anticipated further responses. Your insights and suggestions are not only highly appreciated but also integral to our process, and we stand ready to make any necessary improvements to the paper. If you have any additional questions or require further clarification, please do not hesitate to reach out.

Best Regards,

Submission 4715 Authors

---

### Meta-Review · Area_Chair_dtcq · 2026-01-01

**Summary:**

This paper proposes an end-to-end self-play RL framework to train LLM agents for multi-turn, multi-agent interaction across competitive and cooperative strategic games, with turn-level sum-then-normalize advantage estimation agent-specific advantage normalization. Reviewers agreed the empirical results are compelling and potentially impactful. The main initial concerns were novelty and positioning relative to SPIRAL and RAE and concurrent turn-level methods, confounding between the algorithm and the mixed game curriculum, and (explanation of what drives transfer beyond superficial improvements. The authors provided substantial rebuttal experiments and analyses that generally address these concerns. I recommend accept.

**Reviewer Concerns:**

Addressed:
- Ran the key deconfounding controls experiments, supporting that both the algorithm and mixed curriculum matter.
- Clarified novelty vs SPIRAL/RAE and MT-GRPO.
- Added analysis for why transfer happens and added 8B scaling, and hyperparameter analyses.

Remaining:
- Evidence is still mostly small 2-player games, so broader multi-agent generality claims should be stated cautiously.

**Reviewer Scores:**

YdbT (8): Likely unchanged.

dt93 (6): Unchanged or increase.

3fZ1 (6): Unchanged or increase.

yhYi (4): Unchanged or increase.

---

### Decision · Program_Chairs · 2026-01-26

Accept (Poster)